# Evaluation of statistical climate reconstruction methods based on pseudoproxy experiments using linear and machine learning methods

Zeguo Zhang, Sebastian Wagner, Marlene Klockmann, Eduardo Zorita

Helmholtz Zentrum Hereon, Institute of Coastal Systems, 21502 Geesthacht, Germany

5  *Correspondence to*: Zeguo Zhang (zeguo.zhang@hereon.de)

**Abstract.** Three different climate field reconstruction (CFR) methods are employed to reconstruct spatially resolved North Atlantic-European (NAE) and Northern Hemisphere (NH) summer temperature over the past millennium from proxy records These are tested in the framework of pseudoproxy experiments derived from two climate simulations with comprehensive Earth System Models. Two of these methods are traditional multivariate linear methods (Principal Components Regression, PCR and Canonical Correlation Analysis, CCA), whereas the third method (Bidirectional Long-Short-Term Memory Neural Network, Bi-LSTM) belongs to the category of machine learning methods. In contrast to PCR and CCA, the Bi-LSTM does not need to assume a linear and temporally stable relationships between the underlying proxy network and the target climate field. In addition, Bi-LSTM naturally incorporates information of the serial correlation of the time series. Our working hypothesis is that the Bi-LSTM method will achieve a better reconstruction of the amplitude of past temperature variability. In all tests, the calibration period was set to the observational period, and the validation period was set to the pre-industrial centuries. All three methods tested herein achieve reasonable reconstruction performance on both spatial and temporal scales, with the exception of an overestimation of the interannual variance by PCR, which may be due to overfitting resulting from the rather short length of calibration period and the large number of predictors. Generally, the reconstruction skill is higher in regions with denser proxy coverage, but it is also reasonable high in proxy-free areas due to climate teleconnections. All three CFR methodologies generally tend to more strongly underestimate the variability of spatially averaged temperature indices as more noise is introduced into the pseudoproxies. The Bi-LSTM method tested in our experiments using a limited calibration dataset shows relatively worse reconstruction skills compared to PCR and CCA and, therefore, our working hypothesis that a more complex machine-learning method would provide better reconstructions for temperature fields was not confirmed. In this particular application with pseudoproxies, the implied link between proxies and climate fields is probably close to linear. Yet, a certain degree of reconstruction performance achieved by the nonlinear LSTM method shows that skill can be achieved even when using small samples with limited datasets, which indicates that the Bi-LSTM can be a tool for exploring the suitability of nonlinear methods CFRs especially in small data regimes.

## 1 Introduction

The reconstruction of past climates helps to better understand past climate variability and pose the projected future climate evolution against the backdrop of natural climate variability (Mann and Jones, 2003; Jones and Mann, 2004; Jones et al., 2009; Frank et al., 2010; Schmidt, 2010; Christiansen and Ljungqvist, 2012; Evans et al., 2014; Smerdon and Pollack, 2016; Christiansen and Ljungqvist, 2017). Paleoclimate reconstructions also provide us with a deeper perspective to better understand the effect of external forcing on climate (Hegerl et al., 2006, 2007; Schurer et al., 2013, 2014; Anchukaitis et al., 2012, 2017; Tejedor et al., 2021). However, systematic observational/instrumental climate records are only available starting from the middle of the 19th century, which fails to capture the full spectrum of past climate variations. Consequently, our understanding of climate variations prior to 1850 is mainly based on indirect proxy records (such as tree rings, ice cores, etc. Jones and Mann, 2004). The reconstruction of past climates based on proxy data requires the application of statistical methods to translate the information contained in the proxy records into climate variables such as temperature. These methods add an additional layer of statistical uncertainty and bias to the final reconstruction, in addition to the uncertainties originating in the sparse data coverage and in the presence of non-climatic variability in the proxy records. All these sources of error impact the quality of climate reconstructions. One way to estimate this impact is the test of reconstruction methods in the controlled conditions provided by climate simulations with state-of-the-art Earth System Models. These models provide virtual climate trajectories, which although possibly not completely realistic, are from the model's perspective physically consistent. The skill of the statistical method, the impact of proxy network coverage and of the amount of climate signal present in the proxy records can thus be evaluated in that virtual reality of climate models, once adequate synthetic proxy records are constructed. These tests are generally denoted pseudo-proxy experiments (PPEs; Smerdon, 2012; Gómez-Navarro et al., 2017).

Many scientific studies that employ pseudo-proxies and real proxies have focused on global, hemispheric climate field or climate index reconstructions (Mann et al., 2002, 2005; von Storch et al., 2004; Smerdon, 2012; Michel et al., 2020; Hernández et al., 2020). These studies have identified several deficiencies that are common to most climate reconstructions methods, such as a general tendency to 'regress to the mean', which results in an underestimation of the reconstructed climate variability. This underestimation becomes more evident when the available proxy information becomes of less quality - diminishing the climate signal contained in the proxy records. In addition, sparser networks - shrinking proxy network coverage - may lead to biased reconstructions (Wang et al., 2014; Evans et al., 2014; Amrhein et al., 2020; Po-Chedley et al., 2020). Thus, significant scope still remains for further developing and evaluating climate field reconstructions (CFR) methodologies and in designing methods that are less prone to those common deficiencies (Christiansen and Ljungqvist, 2017).

In the present study, we test a non-linear CFR method that belongs to the machine learning family, a Bidirectional Long-Short-Term Neural Network (Bi-LSTM) and that, to our knowledge, has not been applied to CFR yet. We compare the performance of this method to two well-established classical multi-variate linear regression methods, Principal Component Regression (PCR) and Canonical Correlation Analysis (CCA). Traditional CFRs usually assume linear and temporally stable relationships between the local variables captured by the proxy network and the target climate field. Likewise, the spatial patterns of climate

variability are considered as stationary (Coats et al., 2013; Pyrina et al., 2017; Wang et al., 2014; Smerdon et al., 2016; Yun et al., 2021). However, links between climate fields can be non-linear (Schneider et al., 2018; Dueben and Bauer, 2018; Huntingford et al., 2019; Nadiga, 2020). Nonlinear machine leaning-based CFR methods (for instance, Artificial Neural Networks-ANN) could help capture underlying linear and nonlinear relationships between proxy records and the large-scale

climate more possible (Rasp and Lerch, 2018; Schneider et al., 2018; Rolnick et al., 2019; Huang et al., 2020; Nadiga, 2020; Chattopadhyay et al., 2020; Lindgren et al., 2021). Moreover, machine-learning methods do not necessarily rely on statistical methods to first obtain the principal spatial climate patterns, such as Principal Component Analysis-PCA. The full inherent variability in the original dataset is sequentially and dynamically adjusted and captured with optimized hyper-parameters during the model training process (Goodfellow et al., 2016).

Within the family of machine learning methods, recurrent neural networks (RNN) and Long Short-Term Memory networks (LSTM) are characterized by specifically incorporating the sequential structure of the predictors to estimate the predictand (Bengio et al. 1994). This property makes them promising methods to ameliorate the underestimation of variability that affects many other methods. Our assumption here is that the methods would be able to better capture episodes of larger deviations from the mean, especially those that stretch over several time steps. However, this assumption is not guaranteed

to be realistic in practical situations and needs to be tested. The classical recurrent neural network and Long Short-Term Memory Network can usually only receive and process information from prior forward inference steps. A variant of the LSTM network is the bidirectional Bi-LSTM. It handles information from both forward and backward temporal directions (Graves and Schmidhuber, 2005). It has been demonstrated that the Bi-LSTM model is capable of learning and capturing long-term dependencies from a sequential dataset (Hochreiter and Schmidhuber, 1997) and that it achieves better performance for some

classification and prediction tasks (Su et al., 2021; Biswas and Sinha, 2021; Biswas et al., 2021). Since climate dynamics usually exhibit temporal dependencies, the Bi-LSTM method might learn these dependencies better, which can provide another advantage to capture the time evolution of the reconstructed climate field.

The Bi-LSTM combines two independent LSTMs together, which allows the network to incorporate both backward and forward information for the sequential time series at every time step. Our working hypothesis is, that a more sophisticated type

of RNN could better replicate the past variability, and perhaps even more so for extreme values. Thus, we would like to test whether this property of the Bi-LSTM is useful for paleo climate research in the future based on our experiments, especially by employing only a limited calibration/training dataset that could also be a challenge for training deep neural networks (Najafabadi et al. 2015).

This calibration period, which is usually chosen in the real reconstructions as the observational period (or the overlap period

between observations and proxy records) can represent a challenge not only for a parameter-rich method such as the Bi-LSTM, but also for the usual linear methods. For instance, a global or hemispheric proxy network may span of the order of 100 sites, and a regional proxy network may span a few tenths of sites. If the calibration period spans at most 150 independent time steps, a method like Principal Component Regression, in which one principle component is predicted by the whole proxy network, is rather close to overfitting conditions, especially in a global or hemispheric case. Canonical Correlation Analysis

with a PCA-prefiltering would be much more robust to the potential overfitting if only a few leading PCs are retained in the prefiltering step (sse Methods). Here, we test the methods in our pseudo-proxy experiments in the conditions as they are usually applied in real reconstructions, in which overfitting may be a real risk.

For the sake of completeness, we briefly mention here relevance for our study of the reconstruction methods that combine the assimilation of information from proxy and from climate simulations (Steiger et al., 2014; Carrassi et al., 2018). The family of
data-assimilation methods constrain or modify the spatially complete output of climate simulations conditional on the available locally sparse information provided by proxy records. Therefore, they are not so strongly constrained, in principle, by the assumption that the spatial covariance is stationary over time. Another advantage is that they provide estimation of reconstruction uncertainties in a more straightforward way, especially those methods formally based on a Bayesian framework. On the other hand, the underlying data-assimilation equations do require the estimation of large cross-covariance matrices,
e.g., based on Kalman Filters, and this usually makes necessary the application of some sort of, subjective, regularization of the error-covariance matrices (Harlim, 2017; Janjić et al., 2018). They also might be computationally much more demanding than purely data-driven methods. Considering the replication of the amplitude of past variations, it depends on factors that are independent of the method itself, such as the variance generated by the climate model and also on the inherent uncertainties of the proxy data. Therefore, an under- or overestimation of reconstructed variance cannot be as characterized as a systemic
property of these methods. They have the very important advantage in that they combine all the available information about past climate (simulations, forcings, proxy data) into a powerful tool.

These special characteristics make the comparison with purely data-driven methods more difficult and probably unfair, since data assimilations uses a much larger amount of information from climate simulations. In addition, this use of information from climate simulations compromises one of the main objectives of climate reconstructions, namely the validation of climate
models in climate regimes outside the variations of the observational period. Therefore, the testing of purely data-driven reconstruction methods retains its relevance, despite the availability of more sophisticated data assimilation methods.

In this evaluation of three climate reconstruction methods, we focus on the whole Northern Hemisphere temperature field and on the temperature field of the North Atlantic European region. In the North Atlantic region, the most important mode of temperature variations at longer time series is the Atlantic Multidecadal Variability (AMV). The AMV is sometimes defined
as the decadal variability of the North Atlantic sea-surface temperature, whereas the term Atlantic Multidecadal oscillation (AMO) is reserved for the decadal internal variations (excluding the externally forced variability). Here we focus on the total variability of the North Atlantic SST and define the index of the AMV as decadal filtered surface temperature anomaly over North Atlantic regions 95°W–30°E, 0–70°N, excluding the Mediterranean and Hudson Bay following Knight et al. (2006). It has been shown that AMV is related to many prominent features of regional or even hemispheric multidecadal climate
variability, for example European and North America summer climate variability (Knight et al., 2006; Qasmi et al., 2017). In this context, we test the reconstruction skill for the spatial resolved summer temperature anomalies over Northern Hemisphere-NH (180°W-180°E, 0-90°N) and North Atlantic European region-NAE (60°W-30°E, 0-88°N), as well as for the spatially averaged AMV and NH summer temperature anomalies, calculated from the spatially resolved reconstructed fields. The

reconstruction of mean temperature series could provide a general assessment of the skill to reconstruct extreme temperature
phases (e.g. related to volcanic eruptions or changes in solar activity) serving as benchmarks to test the potential capability of different CFR methods on those anomalies.

## 2 Data and Methods

### 2.1 Data

#### 2.1.1 Proxy data locations

Regarding the networks of real proxies used so far, St. George and Esper (2019) reviewed contemporary studies on previous NH temperature reconstructions based on tree ring proxies (Mann et al., 1998, 2008, 2007, 2009a, 2009b; Emile-Geay et al., 2017). St. George and Esper (2019) concluded that the present-day generation of tree-ring proxy-based reconstructions exhibit high correlations with seasonal hemispheric summer temperatures and display relatively better skill in tracking year-to-year climatic variabilities and decadal fluctuations than former proxy networks, as found by Wilson et al., (2016) and Anchukaitis

et al., (2017). Thus, we test NH summer temperature CFRs employing a pseudo-proxy continental network that is the result of blending two networks: the PAGES2k Consortium (Emile-Geay et al., 2017) multiproxy network, and the climate-tree-ring network of St. George (2014).

In the oceanic realm in the North Atlantic, additional marine proxy records based on mollusc shell bands (Pyrina et al., 2017) have been also used for climate reconstructions. These records, similarly to the dendroclimatological records, are based on

annual growth bands, are annually resolved, and usually represent surface or subsurface water temperature. Therefore, they are technically rather similar to dendroclimatological records. Compelling evidence has already been provided by earlier studies that Atlantic Ocean variability is an important driver of European summer climate variability (Jacobeit et al., 2003; Sutton and Hodson, 2005; Folland et al., 2009). Thus, we also employ an updated proxy network by combining the locations of marine proxies and tree ring proxies (Pyrina et al., 2017; Emile-Geay et al., 2017; Luterbacher et al., 2016) to test the NAE

summer temperature reconstructions.

The pseudoproxies are constructed from the simulated grid-cell summer mean temperature sampled from two climate model simulations over the past millennium (see following subsections). In this context, 11 real proxy locations in the North Atlantic-European region (Pyrina et al., 2017; Emile-Geay et al., 2017; Luterbacher et al., 2016) are selected for regional NAE (60°W-30°E, 0-88°N) PPEs and 48 proxy locations across the Northern Hemisphere are chosen from the PAGES 2k network. The

original Northern Hemisphere PAGES network was trimmed down by removing proxies that may show a combined temperature-moisture response and by selecting only one proxy among those deemed to be too closely located (and thus redundant from the climate model perspective). Specifically, the 48 dendrochronology locations are selected according to Figure 4 of St. George, (2014) which shows the correlation coefficient between the dendroclimatological proxy records and

summer temperature. At most of the retained locations, the correlation between the dendroclimatological record and regional temperature is higher than 0.5.

### 2.1.2 Climate Models

The choice of climate models to run pseudo-experiments will have an impact on the estimation of method skills (Smerdon et al., 2011, 2015; Parsons et al., 2021), since the spatial and temporal cross-correlations between climate variables are usually model dependent. Thus, it is advisable to use several 'numerical laboratories' and employ several comprehensive Earth System Models (ESMs) to evaluate reconstructions methods. Constructing PPEs based on different ESMs will highlight model-based impacts on the reconstructed magnitude and spatial patterns (Smerdon et al., 2011, Smerdon, 2012; Amrhein et al., 2020). Accordingly, in this study two different comprehensive Earth System Models are employed as 'surrogate climate database for setting up PPEs: the Max-Planck-Institute Earth System Model model MPI-ESM-P and the Community Earth System Model CESM.

One of the climate models utilized in our study is the Max-Planck-Institute Earth System model MPI-ESM-P with a spatial horizontal resolution of about 1.9 degree in longitude and 1.9 degree in latitude. The simulation covers the period from 100 BC to 2000 CE. The model MPI-ESM-P consists of the spectral atmospheric model ECHAM6 (Stevens et al., 2013), the ocean model MPI-OM (Jungclaus et al., 2013), the land model JSBACH (Reick et al., 2013) and the bio-geophysical model HAMOCC (Ilyina et al., 2013). The setup of our simulations corresponds to the MPI-ESM-P LR setup in the CMIP5 simulations suite. However, since the present simulations does not belong to the CMIP5 project, the forcings used in this simulation and additional technical details are shown in the Appendix A.

The second climate model is the Community Earth System Model CESM Paleoclimate model from the National Centre for Atmospheric Research (NCAR) (Otto-Bliesner et al., 2016) with a spatial resolution of 2.5 degree in longitude and 1.9 degree in latitude (https://www.cesm.ucar.edu/projects/community-projects/LME/). The CESM simulation extends from 850 CE to 2006 CE using CMIP5 climate forcing reconstructions (Schmidt et al. 2011) and reconstructed forcing for the transient evolution of aerosols, solar irradiance, land use conditions, greenhouse gases, orbital parameters, and volcanic emissions. The atmosphere model employed in CESM is CAM5 (Hurrell et al., 2013), which is a significant advancement of CAM4 (Neale et al., 2013), whereas CCSM4 uses CAM4 as its atmospheric component. The CESM uses the same ocean, land and sea ice models as CCSM4 (Hurrell et al., 2013) does. We use the last one ensemble simulation member 13 from the Last Millennium Ensemble (LME).

## 2.2 Methods

### 2.2.1 Construction of pseudo-proxies

To test the statistical reconstruction methods in the virtual laboratories of climate model simulations, we need records that mimic the statistical properties of real proxy records. The most important properties are their correlation to the local temperature and their location in a proxy network. A third important characteristic is the network size and temporal coverage. The usual method to produce pseudo-proxy records in climate simulations is to sample the simulated temperature at the grid cell that contains the proxy location and contaminate the simulated temperature with added statistical noise, so that the correlations between the original temperature and the contaminated temperature resembles the typical temperature-proxy correlations. The real correlation is of the order of 0.5 or above for good proxy records. This parameter can be modulated in the pseudo-proxy record by the amount of noise added to the simulated temperature, and different proxy networks will help us to reveal how and to what extent degradations of reconstruction skill caused by the amount of non-climatic signals present in the pseudo-proxies.

Ideal pseudo-proxies contain only the temperature signal subsampled from the climate model. We then perturb the ideal pseudo-proxies with Gaussian white noise, and also with red noise for a more realistic noise contamination experiment. We generate two types of pseudoproxies by adding Gaussian white noise and red noise (refer to Pyrina et al., 2017) to the subsampled summer-temperature time series at the tree ring proxy-based locations.

The noise level can be defined using various criteria including signal to noise ratio (SNR), variance of noise (NVAR), and percent of noise by variance (PNV) (Smerdon, 2012; Wang et al., 2014). We employ here the PNV to define the noise level convention. The PNV expresses the ratio between the added noise variance and the total variance of resulting the pseudo-proxy time series. Without loss of generalization we assume that the ideal proxy has unit variance, and thus

$$PNV = NVAR/(1 + NVAR) \qquad (1)$$

Red noise for a specific PNV could be defined by:

$$Red_t = \alpha_1 Red_{t-1} + White_t \qquad (2)$$

where $Red_t$ represents red noise time series, $\alpha_1$ indicates the damping coefficient, here in our study it is equal to 0.4 (Larsen and MacDonald, 1995; Büntgen et al., 2010; Pyrina et al., 2017), and $White_t$ is a random white noise time series correspondingly.

Although individual real proxies contain different amounts of noise (non-climatic variability), we assume here an uniform level of noise throughout the whole pseudo-proxy network. In addition, real proxy records contain temporal gaps, and not all records span the same period. For the sake of simplicity, we assume in our pseudo-proxies network that the data have no temporal gaps and all records cover the whole period of the simulations.

The dataset employed here for constructing the according PPEs database is split into a calibration period that spans 1900-1999AD, and a validation period that spans 850-1899 AD. This calibration period would represent the typical period of

calibration of real proxy records. All the validation statistics of the CFR results are derived against the reconstruction period of 850-1899 AD.

## 2.2.2 Principal component regression

Principal component analysis is employed to construct a few new variables that are a linear combination of the components of the original climate field, and that ideally describe a large part of the total variability. The linear combinations that define the new variables are the eigenvectors of the cross-covariance matrix of the field. Associated to each variable (eigenvector), a principal component time series (scores) describes its temporal variation. In the PCR, the predictands are those scores identified by PCA of the climate field (Hotelling, 1957; Luterbacher et al., 2004; Pyrina et al., 2017). This results in a reduction of dimensionality without losing too much information, and reduces the risk of over-fitting. In the present study, the retained PCs capture at least 90% of the cumulative temporal variance of climate field. After selecting the empirical orthogonal functions-EOFs and principal components-PCs based on the calibration dataset and establishing the desired linear regression relationships between the PCs and the proxy dataset (predictors), the PCs in the validation period are reconstructed using the estimated regression coefficients. The full climate field is then reconstructed by the linear combination of the reconstructed PCs and their corresponding EOFs. A given climate field $x_t$, at time step $t$ can be decomposed as follows:

$$x_{m,t} = \sum_{n=1}^{k} PC_{n,t} \, EOF_{m,n} \tag{3}$$

where $m$ is the grid index of the field, $t$ is the time index, and $k$ denotes the total numbers of retained PCs.

The linear relationship between proxies and targeted climate field is established by the regression equation:

$$PC_{n,t} = \sum_{m=1}^{j} \omega_{n,m} Proxy_{m,t} + \varepsilon \tag{4}$$

where the index $m$ runs over the proxies, $j$ denotes the total numbers of proxies, $\omega$ is the linear function coefficient, and $\varepsilon$ denotes a residual term. The residual could be an unobserved random variable that adds noise to the linear relationship between the dependent variable (PC) and the targeted regressors (proxy or pseudoproxy) and includes all effects on the targeted regressors not related to the dependent variable (Christiansen, 2011).

The $\omega$ parameters are estimated by Ordinary Least Squares. Here, it is assumed that climate sensitive proxies are linearly related with the climate PCs. Based on Eq. (5) using the PCR method, the PCs during the validation interval will be reconstructed assuming that the linear coefficients derived in Eq. (5) are constant in time:

$$\hat{PC}_{n,t} = \sum_{m=1}^{j} \omega_{n,m} Proxy_{m,t} \tag{5}$$

The final reconstructed field $\hat{x}$ will be derived by the linear combination of the reconstructed $\hat{PC}$ with the EOFs derived from the calibration dataset, thereby assuming that the EOF patterns remain constant in time (Gómez-Navarro et al., 2017; Pyrina et al., 2017).

### 2.2.3 Canonical correlation analysis

Canonical Correlation Analysis CCA is also an eigenvector method. Similarly to PCA, CCA decomposes the variance of the fields as a linear combination of spatial patterns and their corresponding amplitude time series. In contrast to PCA, where the

target is to maximize the explained variance with a few new variables, CCA constructs pairs of predictor-predictand variables that maximize the temporal correlation of the corresponding amplitude time series. The pairs of variables are identified by solving an eigenvalue problem that requires the calculation of the inverse of the covariance matrices of each field. These matrices can be pseudo-degenerate (one eigenvalue much smaller than the largest eigenvalue) and therefore the calculation of their inverse is, without regularization, numerically unstable. This regularization can be introduced by first projecting the

original fields onto their leading EOFs (Widmann, 2005; Pyrina et al., 2017). This also reduces the number of degrees of freedom - thus hindering overfitting - and eliminate potential noise variance. After the dimensional transformation, a small number of pairs of patterns with high temporal correlation will be retained. In the present study, the number of retained PCs capture at least 90% cumulative variance of predictand climate field. Then these retained PC time series will be used as input variables of CCA to calculate the canonical correlation patterns (CCPs) and canonical coefficients (CCs) time series for both

proxy and temperature field. The reconstructed climate field can be calculated by a linear combination of the CCPs with CCs for each time step $t$:

$$x_{m,t} = \sum_{n=1}^{l} CC_{n,t}^{field} CCP_{m,n}^{field} \tag{6}$$

$$Proxy_{m,t} = \sum_{n=1}^{l} CC_{n,t}^{proxy} CCP_{m,n}^{proxy} \tag{7}$$

*Proxy* denotes the reconstructed proxy field, and $l$ is the number of CCA pairs. The correlation between each pair CC (proxy,

field) are the canonical correlations, which are the square root of the CCA-eigenvalues. Therefore, once each $CC^{proxy}(t)$ is calculated from the proxy data through the validation period, the corresponding $CC^{field}(t)$ can be easily estimated as proportional to $CC^{proxy}(t)$, since the correlation between the different $CC_n^{proxy}(t)$ is zero. The final reconstruction of target climate field will be derived by linear combination of $CCP^{field}(t)$ and $CC^{field}(t)$, assuming again that the dominant canonical correlation patterns of climate variability are stationary in time.

The CCA method maximizes the correlation that can be attained with a linear change of variables, i.e. with a linear combination of the grid-cell series in each of the two fields. In the following, admittedly artificial, example, the resulting canonical correlation can be very high and yet the reconstruction skill in general can remain low. If one grid cell in each of the two fields are very highly correlated to each other (and assuming here no PCA pre-filtering), CCA will pick those two cells as the first CCA pair (i.e., a pattern in each field with very high loadings only on those cells). The rest of the cells will not contribute to

the CCA pattern. The reconstruction skill will therefore generally be very low in all those cells, despite the canonical correlation being very high. In general, the reconstruction skill will be a monotonic function of the canonical correlation coefficient and the variance explained by the canonical predictand pattern. If the latter is low, the reconstruction skill will be low in large areas of the predictand field, even when the canonical correlation is possibly high.

## 2.2.4 Bidirectional Long Short-term memory neural network

As a non-linear machine learning method, we test here a Bidirectional Long short-term memory neural network (Bi-LSTM). The LSTM networks, in contrast to the more traditional neural networks, also capture the information of the serial co-variability present in the data, and therefore are suitable to tackle data with a temporal structure. These methods are usually applied to the analysis of sequential data, such as speech and time series. The rationale of using these type of networks for climate reconstructions is the hypothesis that a better representation of the serial correlation could ameliorate the aforementioned

underestimation of the past climate variations by most data-driven methods ('regression to the mean', Smerdon, 2012).

The structure of LSTM network is more complex than the structure of a traditional neural network. The LSTM estimates a hidden variable $h(t)$ that encapsulates the state of the system at time $t$. The computation of the new system state at time $t+1$, $h(t+1)$, depends on the value of the predictors at $t+1$ but also on the value of the hidden state at time $t$, $h(t)$. The training of the LSTM can be accomplished sequentially by assimilating the information present in the training data from time steps in the

past of the present time step. In some loose sense, a LSTM network would be the machine-learning equivalent of a linear auto-regressive process.

A Bi-LSTM network, the training of the network is accomplished by feeding it with sequential data iteratively, forwards towards the future and backwards towards the past. Both forward and backward assimilations are processed by two separated LSTM neural layers, which are connected to the same output layer. Figure 1 illustrates the bidirectional structure of the Bi-

LSTM network. Given a set of predictor-predictand variables ($X_t$, $Y_t$), our goal is to train a nonlinear function:

$$\tilde{Y}_t = F(X) \tag{8}$$

where, $\tilde{Y}_t = F(X_t)$ is a close as possible to $Y_t$. The similarity between $\tilde{Y}_t$ and $Y_t$ is defined by a cost function. The structure of this complex non-linear function $F$ is defined as follows:

$$f_t = \sigma\left(W_f[h_{t-1}, x_t] + B_f\right) \tag{9}$$

$$i_t = \sigma(W_i[h_{t-1}, x_t] + B_i) \tag{10}$$

$$A_t = tanh(W_A[h_{t-1}, x_t] + B_A) \tag{11}$$

$$C_t = f_t C_{t-1} + i_t A_t \tag{12}$$

$$o_t = \sigma(W_o[h_{t-1}, x_t] + B_o) \tag{13}$$

$$h_t = o_t tanh(C_t) \tag{14}$$

where $W_f$, $W_i$, $W_A$ and $W_o$ represent several weight matrices and $B_f$, $B_i$ $B_A$ and $B_o$ represent different bias matrices. $\sigma$ is the gate activation function, here we utilize the Rectified Linear Unit function-ReLU (Ramachandran et al., 2017) .

At time step $t-1$, the hidden state of LSTM cell's hidden layer is preserved as $h_{t-1}$, and this vector is combined with the vector of current input variables $X_t$ to obtain the state of the forget gate, $f_t$ (equation 9) , the input gate $i_t$ (equation 10) and the state of

memory cell $A_t$ (equation 11). This memory cell state $A_t$ is linearly combined with the previous state of the cell output $C_{t-1}$ to update the value of its state. The weights of this linear combinations are the states of the forget gate $f_t$ and of the input gate $i_t$ (equation 12). The state of the output gate $o_t$ is calculated from the previous hidden state and the current input variables (equation 13). This output is used to compute the updated hidden state $h_t$ using the state of the cell output $C_t$ (equation 14) (Huang et al., 2020; Chattopadhyay et al., 2020).

In the present application to climate reconstructions, we have a set of input pseudoproxy data $X_t^n = [x_{t-i}, \ldots, x_{t-1}]$ and an output target temperature time series $Y_t^m = [y_{t-i}, \ldots, y_{t-1}]$. The forward LSTM hidden state sequence $\overrightarrow{h_t}$ (note the arrow direction) is calculated employing input information in a positive direction from time $t$-$n$ to time $t$-$1$ iteratively, and for backward LSTM cell, the hidden state sequence $\acute{h}_t$ is computed using the input within a reverse direction from time $t$-$1$ to time $t$-$n$ iteratively. The final outputs from the forward and backward LSTM cells are calculated utilizing the calculation equation (Cui et al., 2018, Jahangir et al., 2020):

$$Y_t = concat\left(\overrightarrow{h_t}, \acute{h}_t\right) \tag{13}$$

where *concat* is the function used to concatenate the two output sequences $\vec{h}$ and $\acute{h}$ (Cui et al., 2018, Jahangir et al., 2020). During training process, the calibration dataset are fed into LSTM cell, and it will map the potential latent relationships (both linear and nonlinear) between input and output variables by updating its weight and threshold matrices. The objective cost function for Bi-LSTM to be minimized during training is the Huber loss that expresses the mismatch between the reconstructed climate field and the 'real' climate field from model simulations. We minimize the loss with gradient descent (Goodfellow et al., 2016). Huber loss has a key advantage of being less sensitive to outlier values:

$$L_\delta\left(Y, f(X)\right) = \begin{cases} \frac{1}{2}\left(Y - f(X)\right)^2 \\ \delta|Y - f(X)| - \frac{1}{2}\delta^2 \end{cases} \tag{14}$$

where $f$ denotes the neural network and the brackets denote the Euclidean norm. The Huber loss function changes from a quadratic to linear when $\delta$ (a positive real number) varies from small to big (Meyer, 2020). Huber loss will approach L2 loss when $\delta$ tends to be 0, and approach L1 when $\delta$ tends to be positive infinity, here we test its value and finally set $\delta$1.35. L2 is the square root of the sum of squared deviations and L1 is the sum of absolute deviations.

The main mechanism of LSTM is that the LSTM block manages to develop a regulated information flow by controlling which proportion of information from the past should be 'remembered' or should be 'forgotten' as time advances. By controlling the regulation of the information flow, LSTM will manage to learn and preserve temporal characteristics and dependencies of the specific time series.

Neural network is generally composed of one input layer, several hidden layers and one output layer. Many hyper-parameters in the neural network usually need to be initialized and tuned for obtaining reasonable results within specific tasks, for instance, activation functions in each layer, objective function for minimizing the loss of the network model, and learning rate for controlling the convergence speed of the network model (Goodfellow et al., 2016). In our specific CFR experiments, we have

explored a range of Bi-LSTM architectures, including different network depths, introducing dropout layers, using different learning rates, and employing different loss functions to provide a more comprehensive evaluation of the Bi-LSTM performance and effectiveness (these tests are shown in Appendix B). These hyper-parameters within Bi-LSTM are finally selected and employed based on our experimental tests (Knerr, et, al. 1990; Kingma and Ba, 2014; Yu, et, al. 2019).

## 3 Results

We evaluate the reconstruction skill of the different methods based on the Pearson correlation coefficient ($cc$) between each target series and the corresponding reconstructed series, and their Standard deviation ratio (SD ratio, SD ratio = $SD_{reconstruction}/SD_{model}$). All the evaluation metrics are calculated in the validation period from 850-1899 AD. High values of derived $cc$ indicate better temporal covariance between target and reconstructed results, a high SD ratio denotes that more variance is preserved in the reconstructions.

### 3.1 North Atlantic-Europe CFRs

Fig. 2 illustrates the CFR results for the North Atlantic-European region employing the 11 ideal-noise-free pseudoproxies based on the three CFR methodologies and the two climate model simulations. When comparing the reconstruction skills across these three CFR methods derived with the same climate model (for example, MPI and CESM correspondingly), the spatial $cc$ patterns calculated between targets and derived reconstructions amongst three CFR methods generally exhibit
similarities. This indicates that all three CFR methods show generally reasonable spatial reconstruction skills (mean $cc$'s over the entire NAE are bigger than 0.4). In addition, $cc$ maps in Fig. 2 show higher values over regions with a denser pseudo-proxy network. This confirms the well-documented tendency amongst different multivariate linear based regression methods for better reconstruction skill in the sub-regions with denser pseudoproxy sampling than in regions with sparser networks (Smerdon, 2010, 2011; Steiger et al., 2014; Evans et al., 2014; Wang et al., 2014). The $cc$ pattern of the nonlinear method Bi-
LSTM is very similar to that of the linear methods, even though the structure of the statistical models is very different. This shows that the nonlinear method employed herein has the similar tendency as linear models to obtain better reconstruction skill over regions with denser proxy sampling.

The picture that emerges from the SD ratio is also very similar for the three methods (Fig. 2). In the regions with a high pseudo proxy density, the SD ratio is high, but outside of the densely sampled areas, all three CFR methods experience a similar degree
of interannual variance underestimation. Appendix C displays the ratio of SD after applying a 30-year filter to the reconstructions and the target fields. The underestimation of variance is larger at these time scale, but the overall conclusion for all three methods remains.

Gaussian white and red noise is constructed and added to the ideal temperature signal of the 11 pseudoproxies subsampled from the MPI and CESM models. The corresponding spatial $cc$ and SD ratio patterns are displayed in Fig. 3 and 4
correspondingly. Compared to reconstructions with ideal pseudo proxies (Fig. 2), a strong degradation of reconstruction skill

amongst all CFR methods occurs over the entire NAE. The reduction in skill is especially profound in the regions where the pseudo-proxy network is denser. Weak reconstruction skill exists over regions where proxies are available and in within their proximity. These noise contamination results shown in Fig. 3 and 4 demonstrate again that the nonlinear method exhibit CFR similarities to the linear methods, whereas, the Bi-LSTM show relatively worse reconstruction skills, with variance

underestimation compared to the other two methods in CESM based PPEs (referring to the spatial SD ratio in Fig. 4).

The ratio of reconstructed to target variance after 30-year low-pass filtering is also larger than for the interannual variance, but otherwise the patterns share the same properties with the ratios of interannual SD (not shown for the sake of brevity).

In general, all three CFR methods exhibit similar reconstruction performance. Specifically, better skills over regions where denser pseudoproxies exist indicates that the spatial covariance patterns learned from the training data (in the 20th century)

are stationary enough to represent the covariance during the reconstruction period over NAE domain It also shows that teleconnection patterns are to some degree localized and do not share considerable amount of covariance outside of the sampled regions.

## 3.2 Northern Hemisphere CFRs

NH summer temperature anomalies reconstructions based on PPEs using three CFR methodologies and the three climate

models are displayed in Fig. 5-7.

Table 1. Skill reconstruction statistics for the Northern Hemisphere mean temperature in the verification period for ideal PPEs. The table shows the result for three CFR methods (PCR, CCA and Bi-LSTM) and two climate models (MPI and CESM). The numbers in parenthesis indicate the skill statistics of white noise and red noise (italics) contaminated PPEs.

| Method | SD Ratio | | $cc$ | |
|---|---|---|---|---|
| | MPI | CESM | MPI | CESM |
| PCR | 0.878(0.904/*0.977*) | 0.874(0.897/*0.913*) | 0.401(0.169/*0.135*) | 0.490(0.216/*0.206*) |
| CCA | 0.603(0.706/*0.694*) | 0.651(0.750/*0.778*) | 0.406(0.165/*0.131*) | 0.507(0.229/*0.218*) |
| Bi-LSTM | 0.710(0.689/*0.669*) | 0.770(0.714/*0.732*) | 0.347(0.145/*0.125*) | 0.462(0.210/*0.191*) |

The spatial $cc$ maps for the ideal PPEs in NH are shown in Fig. 5. Again, all three CFR methodologies yield relatively similar spatial $cc$ patterns of skill for each of the climate models employed here. Skilful reconstructions are again achieved over regions with a denser pseudoproxy network (over North American and Eurasia regions). In addition, relatively high $cc$ values also occur in tropical regions. A relatively high-reconstructed skill is achieved over regions with less or without pseudoproxies, indicating that climate teleconnections between tropics and mid-latitude regions could be responsible for the reconstruction

skill in tropical regions.

All derived CFRs suffer from underestimation of interannual variance, as shown in Fig. 5 and in Table 1, except that the PCR method presents a clearly interannual variance overestimation referring to the specific spatial SD ratio map in Fig. 5. This

overestimation may be impacted by overfitting, since the number of predictors is 47 pseudo-proxies and the calibration period spans 100 time steps. The spatial distributions of the SD ratio also vary between climate models and CFR methodologies. They also are spatially heterogeneous. The CCA method and Bi-LSTM generally preserve more variance over regions with denser pseudoproxies in both CESM and MPI model, and a relatively higher SD ratio appeared in tropical regions within Bi-LSTM based PPEs shown in Fig. 5.

The CCA methodology seems to suffer more strongly from variance losses (see Table 1) over the entire NH compared to PCR and Bi-LSTM.

Considering the general methodological skill, as indicated by the derived spatial mean *cc* and SD ratio values in Table 1, the Bi-LSTM method presents relatively worse performance with lower mean *cc*. The methods PCR and Bi-LSTM generally outperform the CCA methodology with higher mean SD ratio within ideal PPEs.

## 3.3 Spatially variability patterns of the reconstructed fields

In this section, we test the skill of the CFR in replicating the leading spatial patterns of variability, conducting an EOF analysis of the reconstructed temperature fields and compare them with the patterns derived from the target climate simulations. In our PPEs, the dominant patterns of temperature variability are assumed to be stationary. This assumption is also required in real climate reconstructions. Any non-stationarity would be reflected in a loss of reconstruction skill. This type of comparison I srelated to the tests performed by (Yun et al, 2021). In this comparison, the PCA and CCA methods have a clear built-in advantage relative to the Bi-LSTM network, since these two methods operate by design in the space spanned by the leading EOFs of the temperature field. In the case of PCR, these reconstructed fields are a linear combination of the EOF patterns themselves. Therefore, in as much the reconstructed PC series remain uncorrelated, the EOFs of the reconstructed field will be exactly equal to the EOFs of the target climate simulations. Deviations from this behaviour may be caused by the lack of strict orthogonality between the reconstructed PC series caused by the relationship between proxy (predictors) and the PC series (predictands). However, it is reasonably to think that it would not be *a priori* surprising that the EOFs of the PCR-reconstructed fields would be similar to the original EOFs. The case for CCA is theoretically similar, but there are some potentially important points to bear in mind. The CCA patterns, which serve as a basis for the reconstructed field, are linear combinations of the original EOFs. These linear combinations may, for instance, not include the leading EOF of the original field, and thus, the EOFs of the reconstructed field will not replicate the original leading EOF, even if the CCA series can be perfectly reconstructed by the proxy series. The third method Bi-LSTM is in this sense at disadvantage relative to PCR and CCA, since the spatial covariance of the original field is not technically incorporated in its machinery. If the EOF patterns of the reconstructed field resemble the original EOF patterns, this would be an indication that the method itself is able to capture the main covariance patterns of the original field.

In order to have a deeper insight for the reconstruction performance of three CFR methods, we calculated the four leading EOF patterns based on the results from the reconstruction interval, and their proportion of explained variance of the reconstructed field, derived from the three reconstruction methods using the CESM pseudo-proxies. The EOF patterns represented in Figure

8 confirm the suggestions that the temperature reconstructed by the PCR and CCA methods (two lower rows in Figure 8) replicate very closely the three leading patterns. The fourth EOF pattern displays some divergences from the original fourth pattern, but as we will show later, the variance explained by this fourth EOF is already rather low, so that the spatial pattern may be subject to statistical noise. More importantly, the Bi-LSTM method (second row) does produce EOF patterns than

closely resemble the ones derived from the original field. This supports the idea that the method is able to replicate the spatial cross-covariance of the temperature field.

The corresponding spectrum of explained variance is displayed in Figure 9. Here, the percentage of explained variance of each model is calculated as the ratio of the eigenvalue to the total variance of the original field. This is definition is in principle similar to the definition adopted by Yun et al. (2021), but there is one important difference. Yun et al. (2021), according to

their methodological description, calculate the portion of explained variance of each mode as the ratio between the eigenvalue and the total variance of the respective field (either original or reconstructed). This choice could, however, cause a statistical artifact. For instance, when using the PCR regression method, we could choose to reconstruct only the leading EOF pattern. This pattern alone will explain 100% of the reconstructed variance by definition, but this result would be obviously not informative. The choice of the total variance of the original field as reference thus leads to more informative results in general.

The spectra for model simulation and three method-based ideal PPEs in this text are computed as the ratio between each of the first four reconstructed eigenvalues and the cumulative sum of all eigenvalues from the target variable.

### 3.4 An alternative pseudo-proxy network

In this section, we summarize a few additional experiments using the original locations of the PAGES network (Emile-Geay et al., 2017) instead of the filtered network used in previous experiments. In this section, we show only one model test-bed,

for ideal, white-noise and red-noise pseudo-proxies. The results obtained with the MPI-ESM-M model are similar and are, omitted here for the sake of brevity.

The reconstruction skill measured by $cc$ and SD ratio display similar spatial patterns as those obtained with network pre-selected according to the criteria of St. George (2014). As shown in Fig. 10, the derived correlations are generally higher over regions where denser pseudoproxy exits across both ideal and noisy PPEs, and weakly reconstructed correlations appeared

over pseudoproxies-free regions. The PCR method presents a distinct interannual variance overestimation as shown in the specific spatial SD ratio map in Fig. 10 amongst ideal and noisy PPEs, while a clearly interannual variance overestimation also occurs in CCA-based CFRs in the noisy PPEs. A relatively reasonable SD ratio is revealed in tropical regions within Bi-LSTM based PPEs shown in Fig. 10. In general, high reconstruction skills remain over regions where denser pseudoproxy exists based on this additional PAGES 2k pseudoproxy network.

### 3.5 Northern Hemisphere and AMV indices

The evolution of the decadal NH mean temperature anomalies reconstructed by the three CFR methodologies and using pseudoproxies from two models is illustrated in Fig. 11. All indices have been smoothed using a Butter worth low-pass filter

to remove temporal fluctuations shorter than 10 years. The reconstruction performance varies amongst different the CFR methodologies. We will employ the correlation coefficient-*cc*, standard deviation-SD and root mean square error-RMSE as evaluation metrics for NH and AMV indices.

Table 2. *cc*, SD and RMSE (K) during the verification interval for decadal NH mean temperature derived from ideal PPEs. The numbers in parenthesis indicate skill statistics of white and red (italics) noise contaminated PPEs.

| Method | *cc* | | SD | | RMSE | |
|---|---|---|---|---|---|---|
| | MPI | CESM | MPI | CESM | MPI | CESM |
| PCR | 0.880 | 0.871 | 0.821 | 0.763 | 0.086 | 0.072 |
| | (0.632/*0.302*) | (0.532/*0.435*) | (0.806/*0.883*) | (0.502/*0.688*) | (0.143/*0.202*) | (0.122/*0.135*) |
| CCA | 0.882 | 0.853 | 0.704 | 0.560 | 0.091 | 0.086 |
| | (0.664/*0.203*) | (0.536/*0.262*) | (0.647/*0.711*) | (0.464/*0.660*) | (0.135/*0.187*) | (0.122/*0.141*) |
| Bi-lstm | 0.873 | 0.901 | 0.561 | 0.597 | 0.104 | 0.076 |
| | (0.593/*0.351*) | (0.559/*0.394*) | (0.513/*0.540*) | (0.398/*0.470*) | (0.146/*0.173*) | (0.122/*0.133*) |

Table 3. The same as Table 2, but for decadal AMV index

| Method | *cc* | | SD | | RMSE | |
|---|---|---|---|---|---|---|
| | MPI | CESM | MPI | CESM | MPI | CESM |
| PCR | 0.819 | 0.758 | 0.831 | 0.753 | 0.108 | 0.091 |
| | (0.577/*0.336*) | (0.354/*0.429*) | (0.826/*0.961*) | (0.602/*0.837*) | (0.161/*0.213*) | (0.135/*0.139*) |
| CCA | 0.822 | 0.777 | 0.689 | 0.591 | 0.110 | 0.092 |
| | (0.631/*0.288*) | (0.457/*0.424*) | (0.669/*0.744*) | (0.541/*0.766*) | (0.146/*0.200*) | (0.125/*0.136*) |
| Bi-lstm | 0.846 | 0.829 | 0.623 | 0.600 | 0.108 | 0.084 |
| | (0.573/*0.344*) | (0.435/*0.450*) | (0.539/*0.576*) | (0.440/*0.536*) | (0.154/*0.182*) | (0.126/*0.125*) |


The temporal evolution of the original AMV indices (Fig. 12) differs among the simulations, reflecting the different forcings used in each simulation and the model specific contribution of internal variability to the index variations (Wagner and Zorita, 2005; Schmidt et al., 2011). Considering the methodological performance, all three methods generally achieve good AMV index reconstructions when using perfect pseudo-proxies, as shown in each subfigures of Fig. 12 and in Table 3.

The NH and AMV indices derived from more realistic noise contaminated CFRs are shown in Fig. 11 and Fig. 12 correspondingly. The larger noise contamination results in substantial skill deterioration (*cc*, SD and RMSD displayed within brackets in Table 2 and 3). All three methods generally fail to capture the complete variance of the target indices, and the magnitude of strong cooling phases is strongly underestimated.

Fig. 13 illustrates the comparison of Northern Hemisphere indices power spectral density for both, ideal and noise-contaminated PPEs between reconstructions and target models. As indicated in Fig. 13, all three methods generally underestimate the power density, whereas this underestimation is more significant for the noise-contaminated derived PPE.

## 3.6 Probability distributions of reconstructed variables

Even though the three reconstructions methods tend to underestimate the overall variability when using noisy pseudoproxies, an interesting question is their skill in reproducing the probability distributions of the climate indices. In particular, a relevant question is whether the methods are able to capture extreme phases of those indices.

Fig. 14 and 15 display the histogram for the decadal NH mean and AMV indices, respectively. Each subfigure represents the histograms of reconstructed temperature indices across the three methods, compared with the histograms of the target temperature index.

Table 4. Kolmogorov-Smirnov test statistic and p-value for quantifying the histogram distributions between model and reconstructed NH decadal means. Low values of the KS statistic indicate larger similarity between the two distributions. The numbers in parenthesis indicate the KS statistic and p-value of white and red (italics) noise contaminated PPEs.

| Method | KS statistic | | p-value | |
|---|---|---|---|---|
| | MPI | CESM | MPI | CESM |
| PCR | 0.043(0.074/*0.093*) | 0.009(0.193/*0.111*) | 2e-1(6e-3/*2e-4*) | 3e-4(1e-17/*4e-6*) |
| CCA | 0.068(0.081/*0.073*) | 0.171(0.197/*0.130*) | 1e-2(1e-3/*7e-3*) | 6e-14(2e-18/*3e-8*) |
| Bi-lstm | 0.120(0.142/*0.112*) | 0.178(0.241/*0.200*) | 5e-7(9e-10/*3e-6*) | 5e-15(2e-27/*5e-19*) |

Table 5. The same as Table 4, but for AMV index.

| Method | KS statistic | | p-value | |
|---|---|---|---|---|
| | MPI | CESM | MPI | CESM |
| PCR | 0.052(0.050/*0.086*) | 0.101(0.143/*0.085*) | 1e-2(1e-1/*7e-4*) | 3e-5(6e-10/*8e-4*) |
| CCA | 0.082(0.088/*0.083*) | 0.159(0.163/*0.103*) | 1e-3(5e-4/*1e-3*) | 5e-12(1e-12/*2e-5*) |
| Bi-lstm | 0.117(0.154/*0.129*) | 0.172(0.224/*0.191*) | 1e-6(2e-11/*4e-8*) | 4e-14(1e-23/*3e-17*) |

We quantify the distribution similarity between reconstructed and target distributions for both NH and AMV indices using the two-sample Kolmogorov-Smirnov test as a metric (Hodges, 1958) (see Table 4-5). A smaller value of the KS statistic indicates a stronger overall similarity between the two probability distributions. The smallest KS statistic is achieved by the PCR method (see Table 4-5), confirming the impression that the PCR outperforms the other two methods for indices reconstructions in both the ideal and noise contaminated PPEs.

For perfect pseudoproxies, the PCR reconstruction seems to capture the overall target distribution best. It captures the lower tail better than CCA and the upper tail better than CCA and Bi-LSTM. The differences between the methods become smaller for the reconstructions with noisy pseudo proxies, with the PCR still being better than the other two methods (subfigures for the contaminated PPEs in Fig. 14 and 15). The Bi-LSTM performs worst in capturing the lower and upper tails of distribution amongst the three methods, both for the NH mean and the AMV index.

## 505  3.7 Alternative architectures of the Bi-LSTM method

Although the design of machine-learning methods may be guided by the physical considerations, machine-learning methods are still to a large extent a matter of trial and error. The same complexity of the method hinders the disentangling of the causes as to why the methods behave in a certain way. Here, we explore alternative architectures of the Bi-LSTM method to assess the resoluteness of the conclusions drawn from the basic design. We have explored varying network depths (number of layers),
different learning rates, and different cost-functions to optimize the network parameters, among others. A summary of the results is included in the Appendix B.

We could not recognize systematic effects in the skill in this set of different networks designs. The skill varies rather randomly, and probably the identification of optimal network architectures for this specific reconstruction question may not be extrapolated to other applications in paleoclimate. We settled for this application, on a heuristic basis, on an architecture with
2 hidden layers, 4000 hidden nodes, with a learning rate of $10^{-3}$, with the activation function l*eaky relu*, a batchsize of 20 and the Huber loss function.

## 4 Discussion

## 4.1 Nonlinear method performance

Our initial hypothesis was that a more sophisticated model might be able to better capture relationships that are more complex.
For instance, a linear model cannot capture non-linear links outside a narrow range of variations. Artificial neural network is a subset of machine learning method that can be understood as a universal approximator, which can map and approximate any kind of functions by selecting a suitable set of connecting weights and transfer functions (Hornik et al., 1989). Thus, it is reasonable to assume that a better representation of the links between proxy series and climate fields, and thus a better reconstruction performance, might be achieved.
The Bi-LSTM method is the most complex of the three tested in this study. Among them, it is also the one that aims at capturing the serial dependencies. Our hypothesis was that better reconstruction skill could be achieved by the Bi-LSTM method. However, this is not the case in our pseudoproxy experiments. For the spatially resolved NAE fields, the nonlinear Bi-LSTM method achieves a similar skill as the linear PCR and CCA methods, both with ideal and noisy PPEs (see Fig. 2-4).

For spatially resolved NH field, the PCR overestimates the variabilities both in ideal and noisy PPEs (see spatial SD ratio maps
in Fig. 5-7 and mean statistics skills Table 1), and the CCA method shows relatively lower overestimated variance in noisy

PPEs, the Bi-LSTM presents relatively reasonable reconstructions without clearly overestimation both in ideal and noisy PPEs (see Fig. 5-7 and Table 1). Amongst ideal PPEs across two models, the PCR is generally the best method among the three methods, and the nonlinear Bi-LSTM is second best method with higher SD ratio and worse *cc* than CCA method (see Fig. 5-7 and mean skill statistics in Table 1). Whereas, both PCR and CCA exhibit overestimated reconstructions in the amplitude of

climatic variability within noisy PPEs. The Bi-LSTM presents relatively robust reconstructions especially without variance overestimations in noisy PPEs (see Fig. 5-7 and mean skill statistics in Table 1), which may indicate that the LSTM method shows some degree of advantages in reproducing and keeping the general variance within noisy PPEs. The presence of larger noise amplitude causes a deterioration of the Bi-LSTM reconstructions. This may be due to the known sensitivity of this method to the presence of noise. In contrast, the PCR and CCA are less sensitive to the presence of knows and the skill may

even improve in tis settings. A possible reason is the aforementioned overfitting for these two linear methods. The presence of noise ameliorate the collinearity of the proxies given the limited sample size used for training.

For the area-mean indices, all three methods exhibit again generally similar skill. Nevertheless, the Bi-LSTM more strongly underestimates the amplitude of variabilities, and especially over some extreme cooling phases than PCR and CCA. This underestimation is also generally model dependent (see different reconstructed performances in Fig 11-12). In general, the

PCR methods achieved the best performance both in extreme cooling signal capture and indices reconstructions across two models and amongst three methods. The power spectral density plots in Fig. 13 provide a deep insight about these different reconstruction performances in NH temperature indices.

The general inability to capture the cooling extreme signals prior to 20$^{th}$ century indicates that the Bi-LSTM is not good at extrapolating to temperature ranges beyond the training set – a phenomenon that is intrinsic to most ML-based methods.

Therefore, compared with linear methods PCR and CCA, neural network model did not show clear advantages. The performance of the Bi-LSTM might be further improved by optimizing the architecture and parameters of the network, including the type of objective function, type of neural activation function, network optimization function, number of hidden layers, the model-learning rate etc. At this point, it would be quite natural to consider whether the selection/settings of these hyper-parameters in our study is optimal, and also to what extent the reconstruction skill is sensitive to changes in the hyper-

parameters. Nadiga (2020) pointed out that the skill of some machine learning-methods are strongly dependent on these hyper-parameters. Machine learning methods include an extensive range of complexity, and therefore it remains an open issue as to which ML techniques are most or relatively suitable for paleoclimate. It is not clear how the structure of the machine-learning methods can be systematically optimized. At the moment, there is still a considerably amount of 'trial and error' in the design and connection of the neural layers. Here, we have tested the Bi-LSTM network with several different architecture settings,

and finally decided a relatively optimal architecture with two separated hidden layers, and evaluated its performances on CFR experiments, which could be a preliminary try. Our first implementation of the more complex Bi-LSTM does not show superiority in CFRs, at least in our specific experiments, compared to traditional CFR methods, so we would like to draw an assumption that more complicated architecture might not be helpful for CFRs. In addition, a degradation of out-of-sample performance may well be expected when a limited dataset is used to train a neural network model (Najafabadi et al., 2015).

Nevertheless, we would like to point out to other methods, such as an Echo State Network (ESN, Lukosevicius and Jaeger, 2009; Nadiga, 2020) for paleo climate research. Both ESN and LSTM belong to the family of RNN, yet ESN is much simpler than LSTM (Lukosevicius and Jaeger 2009), and has outperformed the RNN methods in other applications (Chattopadhyay et al., 2019; Nadiga, B. 2020). Preliminary pseudo-proxy tests also indicate that this method may improve the deficiencies of the Bi-LSTM. It will be more thoroughly explored in a follow-up study.

Another reason to consider machine-learning methods is the non-linearity of the link between proxies and climate fields. In this particular application with pseudoproxies, the implied link is probably close to linear. However, these can be different on other cases. And it might be the case for more complex problems, (i.e. the reconstruction of proxy-precipiation fields or other modes of natural variability such as the NAO or ENSO). As such, ML methods should not a-priori be excluded from the portfolio of CFR methods leading to more skilful reconstructions of climate.

**4.2 Model and pseudoproxy network dependency**

    The evaluation of the reconstruction skill seems to depend as much on the reconstruction method as on the underlying climate model simulation from which the pseudoproxies were generated. The differences in skill for the same method with different climate model data is of the same order as the differences in skill for the different methods with the same climate model data. The performance of the method does not seem to depend on the domain of the reconstruction. The reconstructions behave

generally similar for the NAE, nevertheless, show some differences in the NH test cases, especially in the derived SD ratio patterns.

    Considering the effects of noise contamination on the methodological performance, both PCR and CCA method exhibit overestimation in the amplitude of reconstructed variability (see SD ratio patters in Fig. 9-10 and mean skills in Table 1). However, all methods suffer from lower correlation coefficients in the more realistic PPEs (white and red noise contaminated

PPEs). The nonlinear Bi-LSTM is more strongly impacted by the noise contamination (Table 1).

    We conclude that noise-contaminated datasets maybe cause obvious overestimations in the amplitude of reconstructed variability for the linear PCR and CCA methods. As some noise signal may deteriorate the reconstructions, while these noise signals may also lead to good reconstructions. The performance of CFR reconstructions is effected by many factors, such as the proxy numbers and its spatial distributions, random noise signal introduced and added to certain important spatial proxy

locations that could have significant effect on the overall spatial reconstruction. For the nonlinear machine learning methods, most of them are very sensitive to external noise. Kalapanidas et al. (2003) and Atla et al. (2011) demonstrated that linear regression can achieve better results than nonlinear methods considering noise sensitivity studies. Moreover, some studies indicated that external interference or noise could damage the ability of neural networks (Heaven, 2019), which may indicate different or more noise level can lead to worse performance for the nonlinear machine-learning method LSTM.

From the perspective of the spatial coverage of the proxy network, the spatial cc and SD ratio patterns (except PCR method) reveal reconstruction skill over the entire NH regions, although this skill is weaker in areas more poorly sampled by the pseudo-proxy network (spatial *cc* patterns in Fig. 5-7). Interestingly, the tropical regions do show some reconstruction skill especially

in the derived reconstructions based on Bi-LSTM (spatial SD ratio patterns in Fig. 5-7), although almost no pseudo-proxies are located in the Tropics. This result indicates the climate teleconnections between tropics and mid-latitude regions could lead to some indirect skill. However, the proxy networks and noise scenarios constructed in the context are certainly not able to mimic/simulate the full range of characteristics completely for climatic proxies in the real world.

## 5 Conclusions

A nonlinear Bi-LSTM neural network method to reconstruct North Atlantic-Europe and Northern Hemisphere temperature fields was tested with climate surrogate data generated by simulations with two different climate models. Compared to the more classical methods of linear Principal Components Regression and Canonical Correlation Analysis, the NAE and NH summer temperature field could be reasonably reconstructed using both linear and nonlinear methodologies referring to spatial *cc* metric. In the relatively larger spatial region-NH temperature field, more discrepancies of reconstructions appeared amongst different climate models and methods based on the derived spatial SD ratio metric. The conclusions drawn from this study can be summarized as follows:

1) In general, all three methods display similar skills when using ideal (noise-free) pseudoproxies, while in the more realistic PPEs (noise contaminated PPEs), both PCR and CCA method exhibit an overestimation on temperature variance preservation, in contrast to the nonlinear Bi-LSTM.

2) The pseudoproxy networks used in this study were mostly located in the extratropical regions with only three proxies in the tropical area. All CFR methodologies produce generally good reconstructions in regions where dense pseudoproxy networks are available. Moreover, teleconnections are explored by these CFR methodologies, leading to some weak spatial reconstruction skills outside of the proxy-sampled regions, for instance the tropical region. The classical linear-based PCR method generally outperforms the Bi-LSTM and CCA method in both spatial and index reconstructions.

3) Here, we could draw a general conclusion that nonlinear artificial neural network method Bi-LSTM employed herein is not superior for CFR reconstructions, at least in our PPEs. In general, Bi-LSTM show worse skill in spatial and temporal CFRs than PCR and CCA, also in capturing extremes. Yet, it is advisable to employ a larger set of nonlinear CFR methods to evaluate different model structures, and further test their performance on CFRs.

## Appendix A

The simulation with the model MPI-ESM-P is not part of the standard CMIP5 simulation suite. In the following, we include additional technical details on this simulation. The MPI simulation was started from the year of 100 BC with restart files from a 500-year spin-down simulation experiments forced with constant external conditions representing the year 100 of BC. After 100 BC, variation in volcanic, solar, orbital, and GHG concentrations are implemented. Land usage was held constant until

850 AD with conditions representing those for year 850 AD. The variation of orbital parameters are calculated after the PMIP3-protocol (Schmidt et al. 2011). The solar activity has been rebuilt on the basis of the reconstruction of Vieira et al. 2011 employing the algorithm and scaling outlined in Schmidt et al. 2011 which corresponds to a difference in short-wave top of the atmosphere insolation of 1.25 Wm-2 (~ 0.1%) between the 2nd half of the 20th century (1950 – 2000) and the Maunder Minimum (1645 – 1715). Variations in greenhouse gas concentrations related to $CO_2$, $N_2O$ and $CH_4$ are after the reconstruction of the PMIP3 protocol – The concentrations were held constant to the values of year 1 AD between 100 BC and 1 AD because the law Dome records does not extend beyond year 1 AD. After 1850 AD also a reconstructed aerosol loading after Stine et al. 2018 were employed to account for transient anthropogenic aerosol emissions. The extension and reconstruction of the volcanic forcing is related to a rescaling of the newly available Sigl et al. (2015) dataset to the reconstruction of Crowley and Unterman (2013). The large volcanoes for different latitudinal bands are rescaled according to sulfate concentrations and eventually the Crowley algorithm was applied to yield aerosol optical depths and effective radius for four latitudinal bands separated by 30°.

**Appendix B**

We have explored a range of Bi-LSTM architectures, including employing different network depths, introducing dropout layers, using different learning rates, and employing different loss functions to provide a more comprehensive evaluation of the Bi-LSTM performance and effectiveness. Table 1B-6B present reconstruction statistics skill for the spatial North Hemisphere mean temperature in the verification period for ideal PPEs based on CESM using different architecture settings of Bi-LSTM method. In our PPE tests on paleo CFRs, it seems that in this case we could not univocally identify optimal neural network structure that could universally outperform all others. And the final Bi-LSTM architecture employed in our CFR experiments was finally determined with 2 hidden layers with 4000 hidden nodes, learning rate is $10^{-3}$, activation function is leaky relu, batchsize is 20 and Huber loss function.

Table 1B. Different loss function conditioned on other parameters fixed (2 hidden layers with 4000 hidden nodes, learning rate is $10^{-3}$, activation function is leaky relu, batchsize is 20)

| Loss functions | cc | SD Ratio |
|---|---|---|
| MAE | 0.483 | 0.670 |
| MAPE | 0.124 | 0.050 |
| MSE | 0.465 | 0.759 |
| Huber | 0.462 | 0.770 |

MAE: mean absolutely error, MAPE: mean absolutely percentage error, MSE: mean square error, Huber: Huber loss

Table 2B. Different learning rate using Huber loss, and with the rest parameters fixed as in Table 1B

| Learning rates | cc | SD Ratio |
|---|---|---|
| 1e-1 | -7e-3 | 1e7 |
| 1e-4 | 0.462 | 0.770 |
| 1e-6 | 0.462 | 0.675 |
| 1e-8 | 0.012 | 0.271 |

655

Table 3B. Different activation functions with the rest parameters fixed as in Table 1B

| Activation function | cc | SD Ratio |
|---|---|---|
| ReLU | 0.505 | 0.566 |
| Leaky ReLU | 0.462 | 0.770 |
| ELU | 0.529 | 0.617 |
| PReLU | 0.509 | 0.544 |

Table 4B. Different hidden layer number with the rest parameters fixed as in Table 1B

| Number of layers | cc | SD Ratio |
|---|---|---|
| 1 | 0.508 | 0.733 |
| 2 | 0.462 | 0.770 |
| 4 | 0.442 | 0.603 |
| 6 | 0.335 | 0.411 |

Table 5B. Different hidden node numbers in each layer with the rest parameters fixed as in Table 1B

| Number of hidden nodes | cc | SD Ratio |
|---|---|---|
| 200 | 0.479 | 0.620 |
| 1000 | 0.502 | 0.692 |
| 2000 | 0.503 | 0.711 |
| 4000 | 0.462 | 0.770 |

Table 6B. With and without dropout layers conditioned on the rest parameters are fixed as in Table 1B

| Dropout | cc | SD Ratio |
|---|---|---|
| Dropout | 0.462 | 0.770 |
| Non-dropout | 0.467 | 0.760 |

**Appendix C**

Appendix C displays the SD ratios for ideal pseudo-proxies after filtering the reconstructed and target fields with a 30-year low pass filter. At these time scale, the SD ratio is again lower than for the interannual variance.

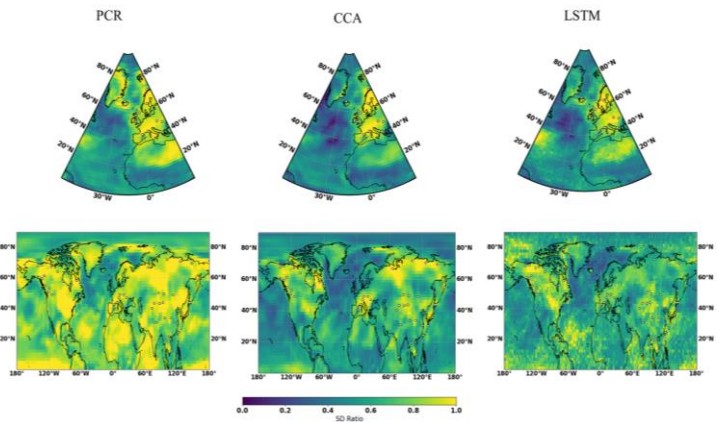

Figure 1C: 30year filtered SD ratio pattern using Ideal-PPEs based on MPI model over validation period 850-1899 for NAE (upper row) and NH (lower row)

**Data availability**

The MPI-ESM-P model output that was employed for this study is available upon request from the authors S.W or E.Z. And from the Paleoclimatology data repository of the National and Oceanic and Atmospheric Administration in the U.S.A (https://www.ncei.noaa.gov/products/paleoclimatology [Note for the reviewers and editor: these data will be uploaded after publication]). The CESM model data can be downloaded from https://www.cesm.ucar.edu/projects/community-projects/LME/.

## Author contributions

The analysis was performed by ZZ with the consultation of SW, MK and EZ. ZZ prepared the paper with contributions from all co-authors.

## Competing interests

The authors declare that they have no conflict of interest.

## Acknowledgements

The authors thank the MPI-ESM modelling groups participating in the CMIP5 initiative for providing their data and the CESM modelling group for making their data available. The authors thank the editor and the referees for constructive suggestions that have improved the content and presentation of this article.

## Financial support

This work is funded by the China Scholarship Council (no. 201806570017), and is part of the project Reduced Complexity Models (Redmod), funded by the Helmholtz Association through its Incubator program.

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

ndex.

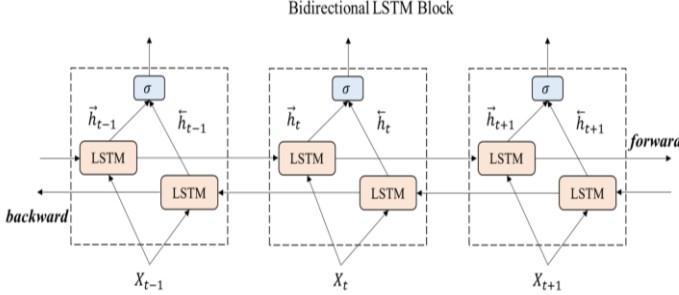


Figure 1: the bidirectional structure of the Bi-LSTM network.

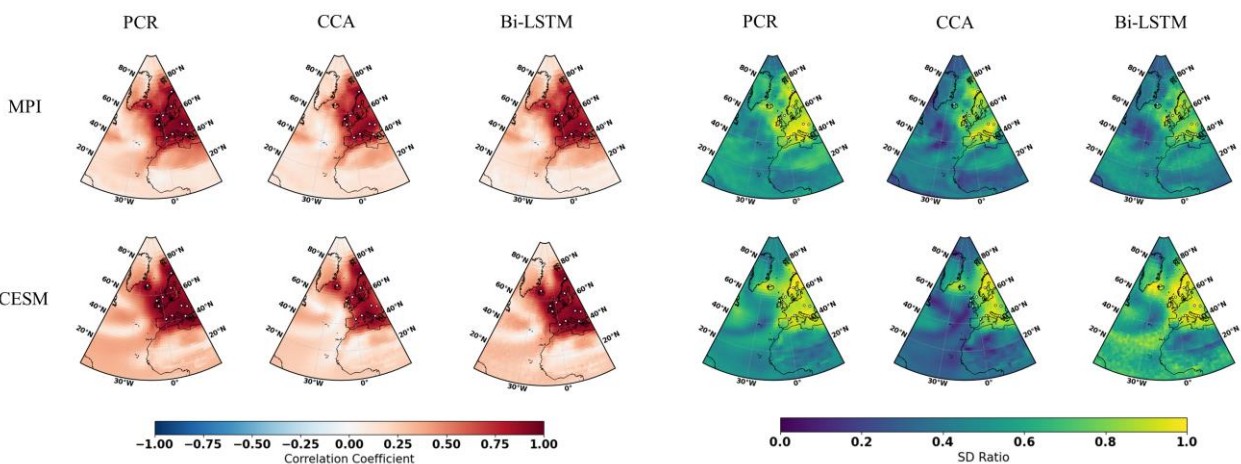

Figure 2: NAE Reconstruction results of CFR methods (including PCR, CCA, Bi-LTSM : Bidirectional long short term
memory neural networks) using MPI and CESM numerical simulation as target temperature field, all the CFR methods employ
the same proxy network with full 11 ideal pseudoproxies which span the same reconstruction period from 850-1899 AD. The
employed pseudoproxies geolocations are show in white circles in all the sub-figures; CC is Correlation Coefficient and SD
represents Standard Deviation Ratio. The employed pseudoproxies' geolocation is shown as white circles in all the sub-figures.

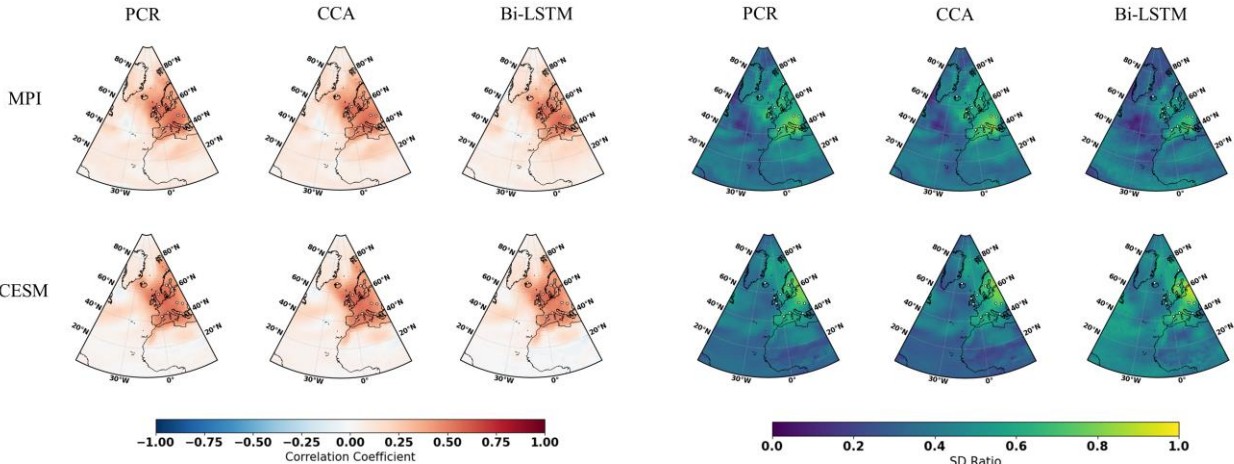

Figure 3: the same as Figure 2, but for employing the full 11 pseudoproxies network with white noise contamination.

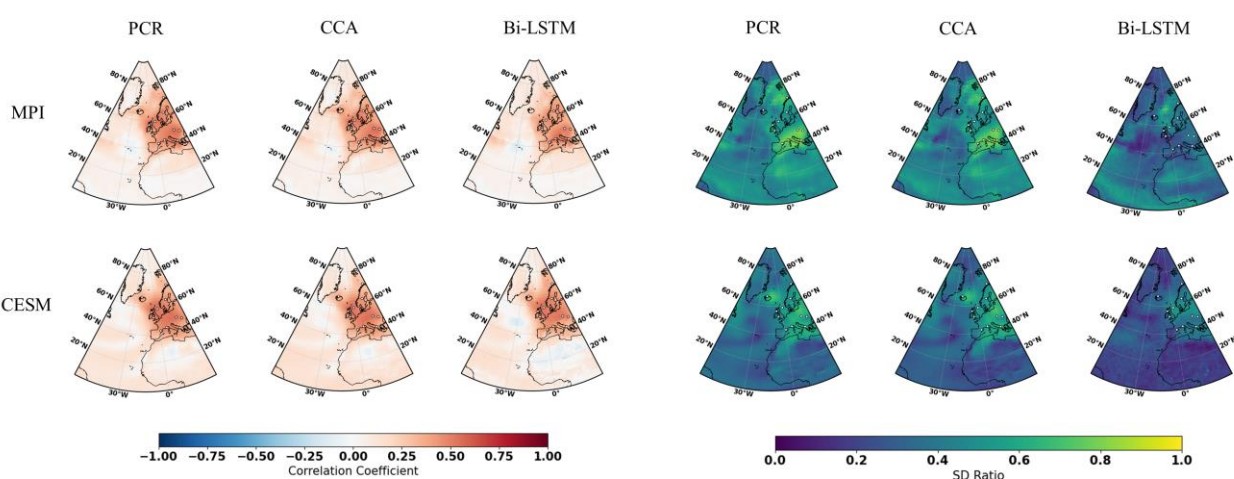

Figure 4: the same as Figure 2, but for employing the full 11 pseudoproxies network with red noise contamination.


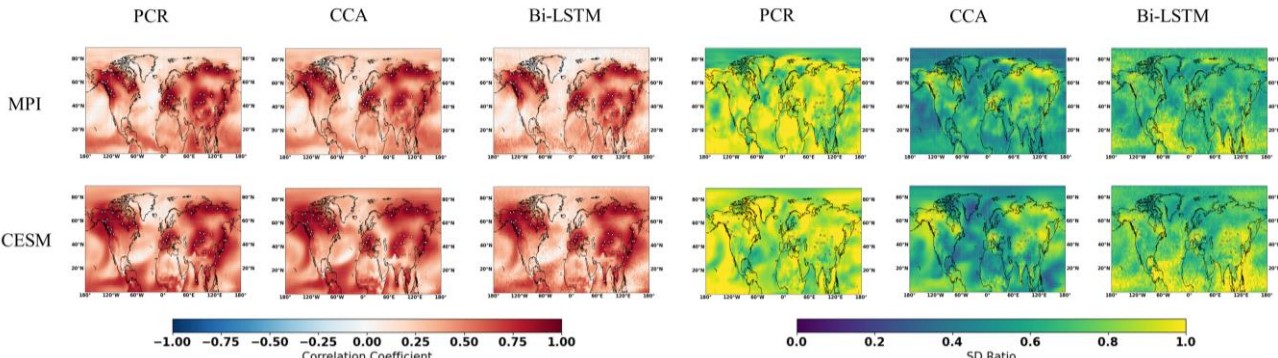

Figure 5: NH Reconstruction results of CFR methods (including PCR, CCA, Bi-LSTM: Bidirectional long short term memory neural networks) using MPI and CESM numerical simulation as target temperature field, all the CFR methods employ the same proxy network with full 48 ideal pseudoproxies which span the same reconstruction period from 850-1899AD. The employed pseudoproxies geolocations based on TRW are shown in white circles in all the sub-figures; CC is Correlation Coefficient and SD represents Standard Deviation Ratio. The employed pseudoproxies' geolocation is shown as white circles in all the sub-figures.

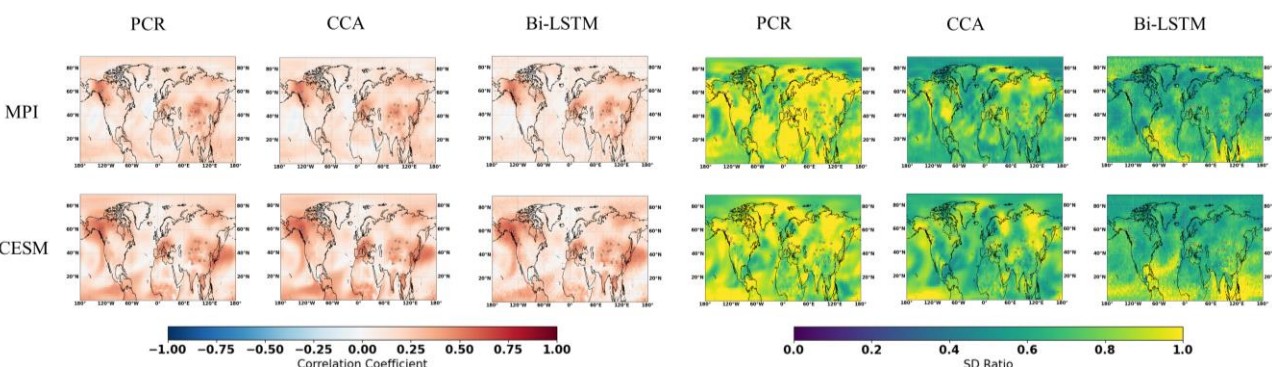

Figure 6: the same as Figure 5, but for employing the full 48 pseudoproxies network with white noise contamination.

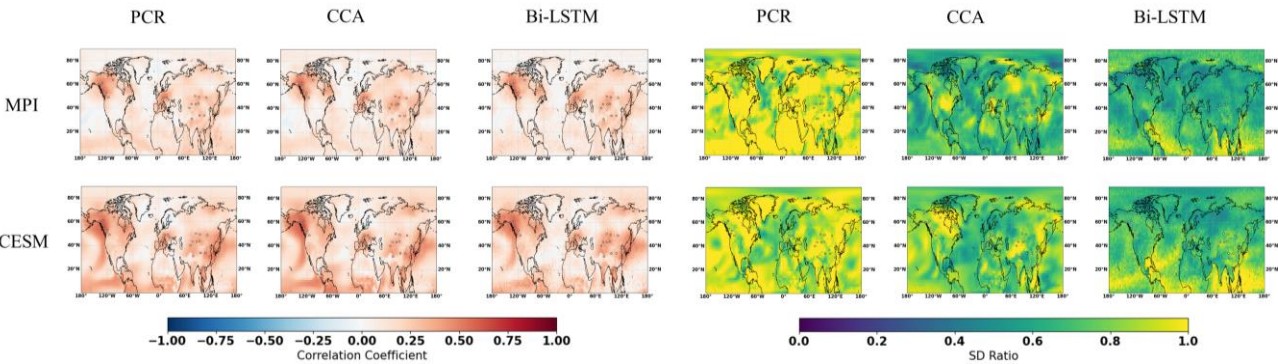

Figure 7: the same as Figure 5, but for employing the full 48 pseudoproxies network with red noise contamination.

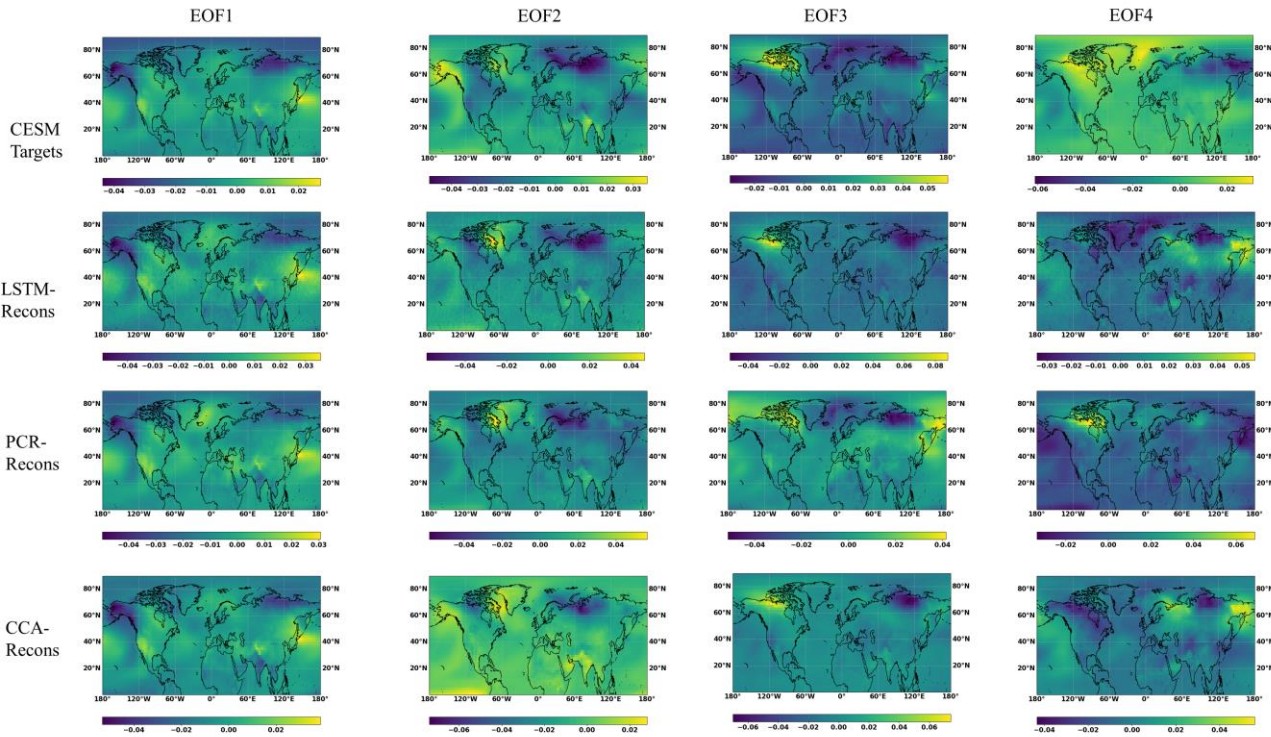

Figure 8: First four EOF patterns of the temperature field derived from for CESM target and derived from the temperature field reconstructed by the three methods-based ideal PPEs reconstructions

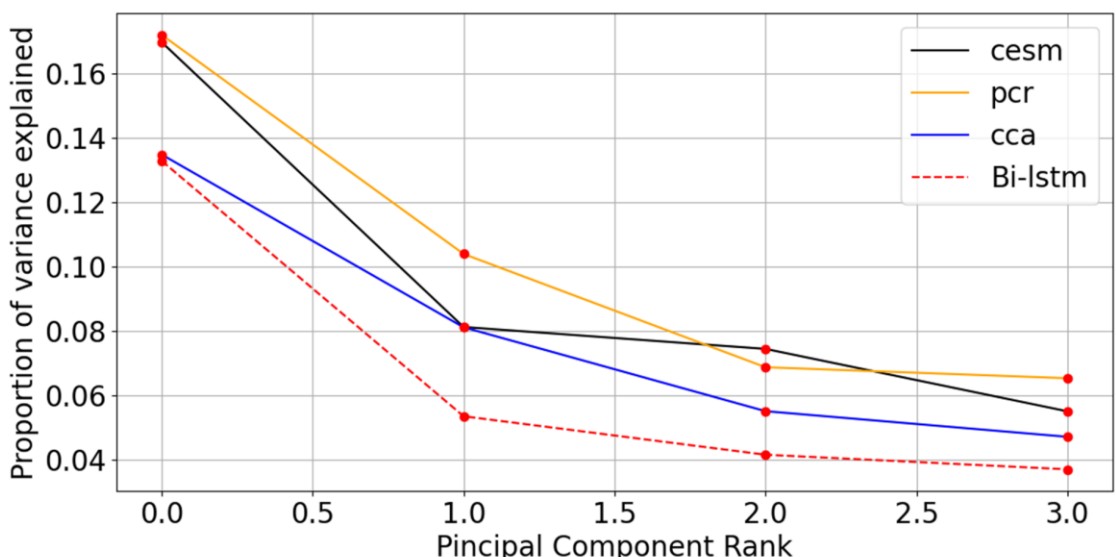

Figure 9: Eigenvalue spectra for CESM simulation and three method reconstructions: the spectra for CESM simulation and three method-based ideal PPEs are computed as the ratio between each of the first four reconstructed eigenvalues and the cumulative sum of all eigenvalues from the target CESM model

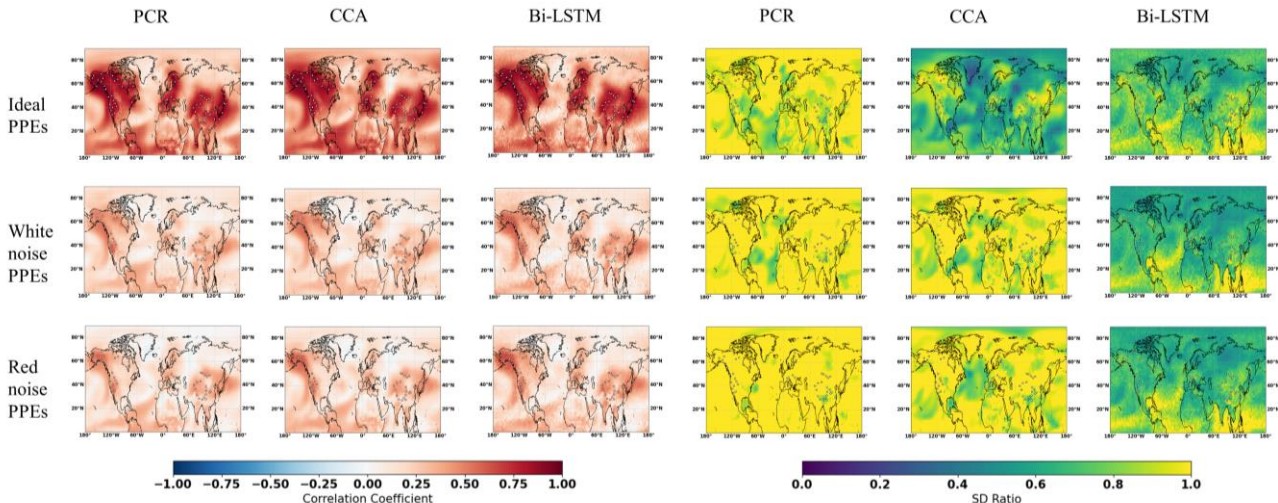

Figure 10: Summary of the pseudo-reconstructions derived from the CESM model-based pseudo-proxies using the original PAGES proxy network. The panels display the maps of the temporal correlation coefficients at the grid-cell level (*cc*) and the ratio of standard deviations (SD ratio) between reconstructed and target temperature field

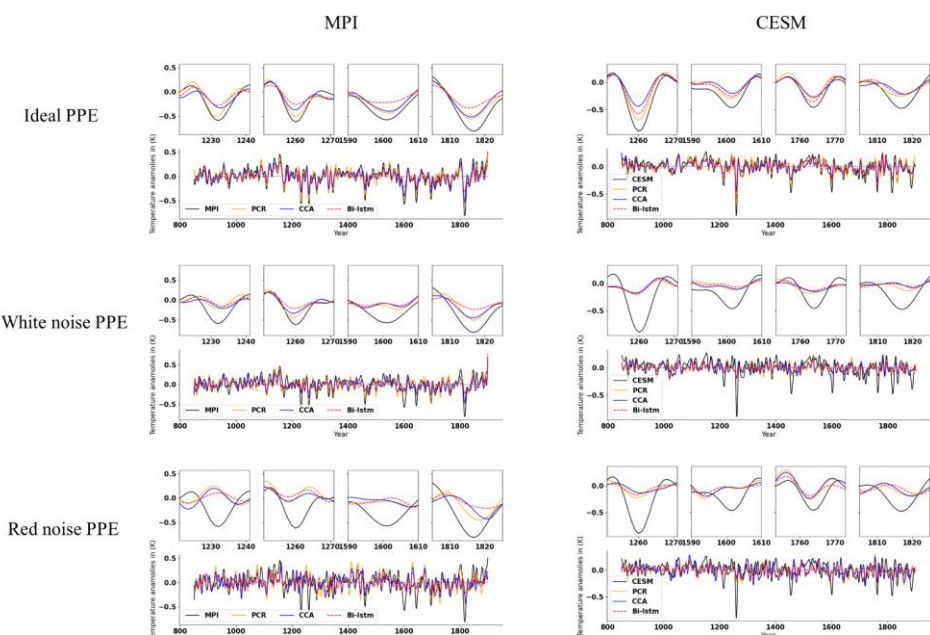


Figure 11: mean time series evolution of the validated reconstructions for NH summer temperature anomaly using full 48 pseudoproxies based on PCR, CCA, Bi-LSTM CFR methods. All time series have been smoothed using a butter worth low-pass filter to remove temporal fluctuations less than 10 years. MPI and CESM represent MPI/CESM model simulated 'true' climatology. We selected several reconstructed extreme cooling period with a shorter interval (each 10 years are selected before and after the specific extreme cooling year) and plotted them above each entire reconstruction means amongst models and methods.

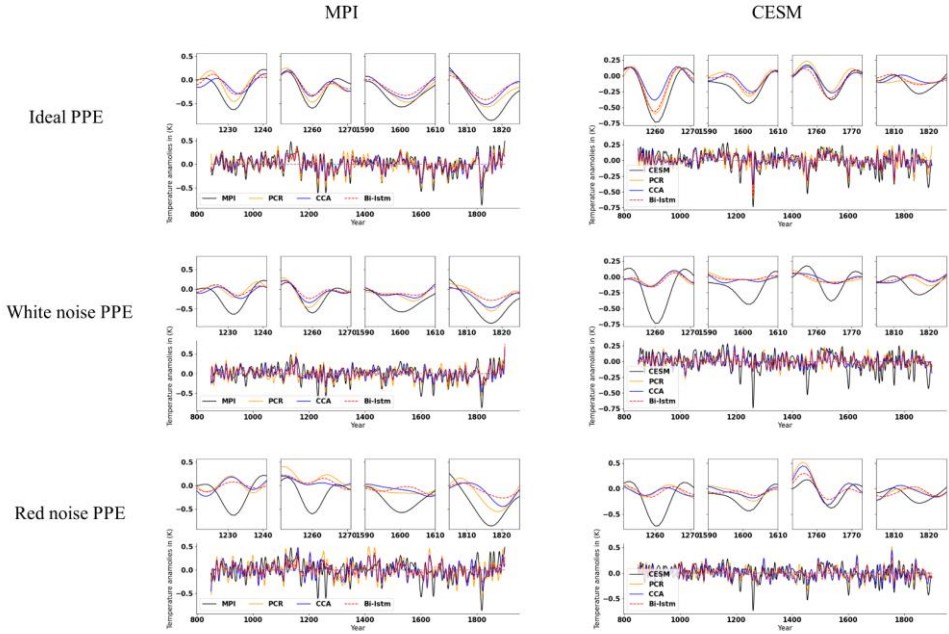

Figure 12: The same as Figure 11, but for Atlantic Multidecadal Variability (AMV) index.

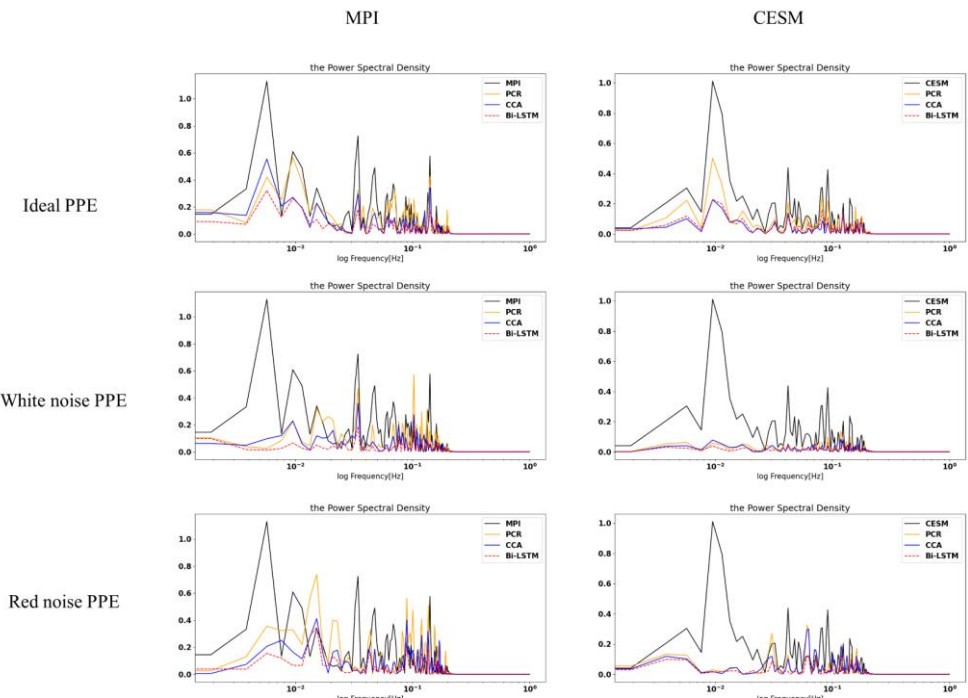

Figure 13: North Hemisphere indices power spectral density.

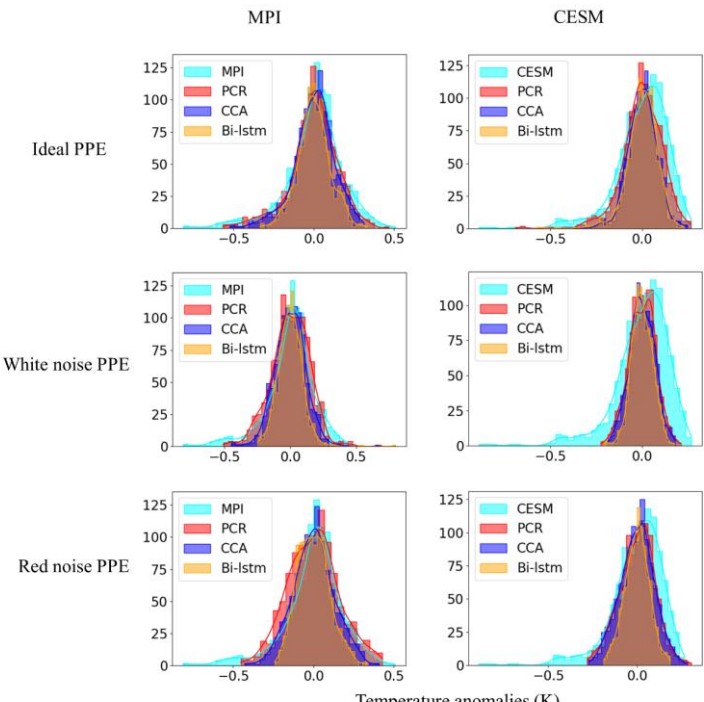

Figure 14: Histogram for decadal filtered NH mean index. The x axis denotes temperature anomaly values, and y axis is the number of data in each bin. Totally 30 bins are selected to plot each of the histogram.

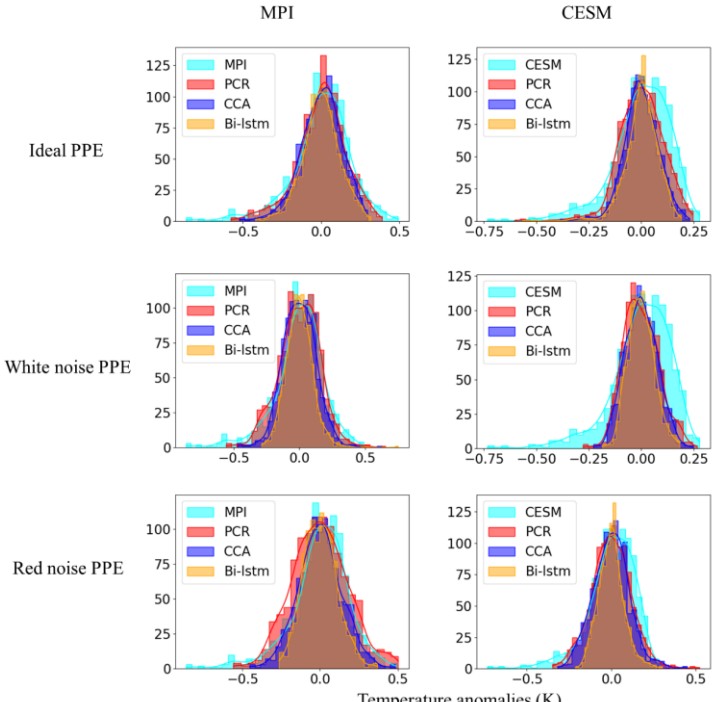


Figure 15: The same as Figure14, but for decadal filtered AMV index.

