# Peer review of "Evaluation of statistical climate reconstruction methods based on pseudoproxy experiments using linear and machine learning methods"

_Climate of the Past, 2022_

## Referee Comment (RC1)

**Comment on CP-2022-5**

**1 General comments**

This paper compares the reconstructive skill of three climate field reconstruction (CFR) methods in an array of pseudoproxy experiments. The authors compare the traditional approaches of principal component regression (PCR) and canonical correlation analysis (CCA) against a proposed Bi-directional Long Short Term Memory (Bi-LSTM) neural network method. Results show that PCR tends to out perform CCA and the Bi-LSTM in reconstruction NAE and NH summer temperature field. However, the Bi-LSTM shows some promise due to its improved ability to estimate the lower tails of the distribution.

I think that the application of LSTMs for CFR an interesting idea, but I do not think this paper goes far enough in its analyses to draw strong conclusions about its effectiveness. Only one LSTM architecture (with varying widths) is presented, while it is customary in deep learning research to evaluate a range of architectures and report the results for a collection of models. Additionally, by only comparing against PCR and CCA, the paper omits some potentially interesting comparisons against more sophisticated techniques like Data Assimilation (DA), which can improve over PCR in pseudoproxy experiments (Steiger et al., 2014). Finally, analysis of the results is largely descriptive of the plots, but does not delve deeply into possible explanations. For instance, its not clear why might the proposed Bi-LSTM model behave so similarly to PCR or why the Bi-LSTM captures the low end of the distribution better than PCR.

Overall, I think a deeper exploration of Bi-LSTM architecture settings, comparisons against DA methods, and more climatological insight into the results are necessary to make the results compelling and generalizable.

**2 Specific comments**

1. I'm still unclear as to whether this article is proposing Bi-LSTMs as a viable alternative to traditional statistical methods or whether its trying to show that they don't

work well enough. If this article is intended to propose using Bi-LSTMs (or to refute their use) then this stance needs to be made more clear up front and in the results. Right now the article presents the results as a neutral "comparison of methods", but this makes it difficult to reach a conclusion about the proposed method.

2. Followup to Q1 – If the article is proposing (or refuting) Bi-LSTMs then it needs a more comprehensive evaluation of the Bi-LSTM model. The results in Appendix B are a good start, but it would have been nice to see how varying the depth of the network, using dropout, weight decay, or other regularization techniques, or varying the learning rates (or using a scheduler), number of epochs, and other aspects of the training procedure such as the loss function effect generalization. From this, a reader could draw broader conclusions about the effectiveness of Bi-LSTM models rather than the effectiveness of the single model presented.

3. The statement "The reconstruction of mean temperature series could provide a general assessment of the skill to reconstruct extreme temperature phases" needs either a citation or experimental results in Section 3. I think extremal behavior could be quite different than behavior near the mean? It would be interesting to see if the Bi-LSTM can model quantiles of the distribution better than PCR or CCA.

4. What is the rationale for training on 850-1425 and then testing on the later period of 1426-2000 (where did 2000-2005 go?), rather than the reverse as in Steiger et al. (2014)? I think that in a paleoclimate experiment we would be more interested in the performance of our method in the relative past, rather than the relative future. Would the performance of different methods change with the temporal order of training and testing?

5. In practice, the Bi-LSTM would need to be trained on real proxies and real observations, which would limit the training period to 1850 onwards. Will this be enough observations, over a long enough time horizon, to train an LSTM model? Comparisons between the various methods under this limited data setting would be helpful. Also, how will you account for the significant covariate shift between the post-industrial and pre-industrial periods?

6. Lines 225-235 seem to motivate including temporal correlation in a model more generally, rather than LSTMs specifically. Since methods like Data Assimilation already model time varying processes, what is the potential benefit of the LSTM? This section

should contain a more clear and comprehensive justification of the LSTM to motivate it over existing time series techniques.

7. Comparing Table 1 and 2, why is SD ratio replaced with RMSE, particularly since RMSE was not mentioned as a comparison metric in the beginning of Section 3. Also, RMSE needs to be defined or at spelled out once.

8. Line 156 states that "We then perturb the ideal pseudo-proxies with Gaussian white noise ... with signal-to-noise ratio (SNR) values of 0.25, 0.5 and 1", but then later on line 306 it states "More realistic pseudo-proxies are those containing 80% Gaussian white noise contamination.", and it would seem that the 80% contamination is used in all of the experiments. How is this 80% number connected the previously stated SNR values?

9. On line 395 – "The Bi-LSTM is able to capture periods of extreme cooling better than the other two methods but strongly underestimates the recent warming trend." Is it possible the LSTM is just biased towards colder temperatures?

10. The figures need to be referenced more heavily in the text. Statements such as "In addition, cc maps show higher values over regions where more pseudoproxies are located." and "The Bi-LSTM and PCR methods exhibit relatively consistent patterns with similar SD ratios" seem to refer to the content of a plot. Without an explicit reference though its hard to follow.

11. I think section 2.2.1. Construction of pseudo-proxies should be grouped in with the Data section 2.1, rather than the Methods section 2.2.

**3   Technical corrections**

There are many grammatical mistakes and informal statements in this paper. It would be good if the authors could thoroughly proofread the paper once more and correct them. I list a few examples here:

1. Line 26 – "which hinders to capture" should be "which fails to capture"

2. Line 27 – "in earlier centuries" is too informal and needs to be specified. Do you mean prior to 1850?

3. Line 27 – "(such as tree rings, ice cores), etc." should be "(such as tree rings, ice cores, etc.)" and should have a citation.

4. Line 75 – "over NH and NAE" these acronyms need to be defined explicitly.

5. Line 98 – remove " Besides,"

6. Line 136 – "labelled past1000 and historical and labelled r1i1p1" makes it sounds arbitrary. I think you mean you "combined past1000 and historical simulations from ensemble member r1i1p1"

7. Line 266 – "Object function" should be objective function. Similarly on line 268 – "Object loss" should be "The loss" or just "We minimize the loss with gradient descent"

8. Line 295 – "all the spatial cc patterns exhibit similarities" this is too informal.

9. Line 324 – "These regions barely contain any pseudoproxies," also too informal.

**References**

Steiger, N. J., Hakim, G. J., Steig, E. J., Battisti, D. S., and Roe, G. H. (2014). Assimilation of time-averaged pseudoproxies for climate reconstruction. *Journal of Climate*, 27(1):426–441.

---

## Author Comment (AC1)

**1. General comments**

**A**: We would like to thank Referee 1 for their detailed and constructive comments.

Regarding their general comments, we agree that it would be beneficial to deepen our exploration of the LSTM sensitivity to different architectures and tunable parameters, as is common for deep learning applications. In the revised version, we will evaluate the LSTM methodology using a range of architectures and explore more deeply the performance of LSTM (see also answer to Specific Comment 2). We will also put our results in context with paleo-data assimilation methods, and explore the reasons why the Bi-LSTM achieves similar reconstruction skill as PCR. We will also take care to add more climatological insight to our analysis.

In the following responses, we specify how we plan to address those points.

**2. Specific comments**

*1. I'm still unclear as to whether this article is proposing Bi-LSTMs as a viable alternative to traditional statistical methods or whether its trying to show that they don't work well enough. If this article is intended to propose using Bi-LSTMs (or to refute their use) then this stance needs to be made more clear up front and in the results. Right now the article presents the results as a neutral comparison of methods", but this makes it difficult to reach a conclusion about the proposed method.*

**A**: The revised version will more broadly explore the Bi-LSTM method, which will allow us to achieve more robust conclusions on the Bi-LSTM. The results achieved so far indicate that the method is not superior to the traditional linear methods, with the possible exception of the replication of cold extremes, but the planned deeper exploration with a range of architectures may modulate this conclusion.

*2 Follow up to Q1-If the article is proposing (or refuting) Bi-LSTMs then it needs a more comprehensive evaluation of the Bi-LSTM model. The results in Appendix B are a good start, but it would have been nice to see how varying the depth of the network, using dropout, weight decay, or other regularization techniques, or varying the learning rates (or using a scheduler), number of epochs, and other aspects of the training procedure such as the loss function effect generalization. From this, a reader could draw broader conclusions about the effectiveness of Bi-LSTM models rather than the effectiveness of the single model presented.*

**A**: We will explore a range of architectures, different network depths, introducing dropout layers, using different learning rates, and employing different loss functions to provide a more comprehensive evaluation of the Bi-LSTM performance and effectiveness. Those results will be rather detailed and - depending on the outcome - we will need to take care not to overload the manuscript with figures and tables of sensitivity studies. We therefore plan to include most of these results in an appendix, and retain in the main manuscript only those results/configurations that help support the main conclusions on the Bi-LSTM method.

*3 The statement "The reconstruction of mean temperature series could provide a general assessment of the skill to reconstruct extreme temperature phases" needs either a citation or experimental results in Section 3. I think extremal behavior could be quite different than behavior near the mean? It would be interesting to see if the Bi-LSTM can model quantiles of the distribution better than PCR or CCA.*

**A**: This sentence was meant to justify the consideration of the Bi-LSTM method in the first place. This method, in contrast to the usual set-up with the traditional PCR and CCA methods, naturally incorporates the serial correlations of the inputs. The working hypothesis is that this could improve the reconstruction of extremes, or at least provide a different reconstruction skill than for the mean values. Our objective was

to test this potential difference. We will expand in the revised version the justification of the use of the Bi-LSTM method and the analysis of the skill of all three methods for the specific replication of extremes. The evaluation of the replication of extremes will also need to address not only the amplitude of spatial averages or of the indices, but also the spatial patterns that occur during those extremes. For instance, it will be interesting to show whether deficiencies in the replication of the amplitude of extreme indices or spatial averages is brought about by a general overestimation of the extreme amplitudes or by an incorrect replication of the sign of spatially resolved anomalies, which would cause a subdued extreme after spatial averaging.

*4 What is the rationale for training on 850-1425 and then testing on the later period of 1426-2000 (where did 2000-2005 go?), rather than the reverse as in Steiger et al. (2014)? I think that in a paleoclimate experiment we would be more interested in the performance of our method in the relative past, rather than the relative future. Would the performance of different methods change with the temporal order of training and testing?*

**A**: We agree and will replace the calibration period with the 20 century, and use the rest of the past last millennium time period as validation (also in response to a comment by Reviewer 2). This will provide an even more realistic test of the methods and also address the next comment.

*5 In practice, the Bi-LSTM would need to be trained on real proxies and real observations, which would limit the training period to 1850 onwards. Will this be enough observations, over a long enough time horizon, to train an LSTM model? Comparisons between the various methods under this limited data setting would be helpful. Also, how will you account for the significant covariate shift between the post-industrial land pre-industrial periods?*

**A**: This is a relevant question, which partially explains our unusual choice of the calibration period. We agree with the reviewer that our choice of calibration period cannot be implemented in a real application. We wanted, however, to ameliorate the problem of the estimation of the covariances when using a small (and anthropogenically contaminated) sample size. This would have allowed us to better identify the methodological differences per se. In view of the opinion of the reviewers, we will now replace our choice of the calibration period and use the 20th century, as indicated in the response to the previous comment. We will do PPEs limiting, for example, the training period to 1900 onwards to check its performance. Degradation of performance may well be expected when a limited dataset is used to train a neural network model (Najafabadi et al., 2015). Also splitting data inappropriately might cause unexpected effects on the general performance of one neural network model. In typical neural network tests in the ML context, one splits the data at random for most tasks but available data in the paleoclimate context is of course not random. As the reviewer rightly states, the data might contain trends or shifts in covariance in time – these could result from changes in the way the data was gathered or from varying choices over what information to collect (Riley, 2019) as well as from changes in climate conditions. This is a problem that is inherent to most data-driven climate reconstruction methods, which assumes that the covariance patterns learned from the training data (usually in the 20th century) are stationary enough to also represent the covariance during the reconstruction period (as we have mentioned also in ll.196-197 and ll.220-221 for the PCA and CCA). One objective of the pseudo-proxy experiments is precisely to test the consequences of this assumption.

*6 Lines 225-235 seem to motivate including temporal correlation in a model more generally, rather than LSTMs specially. Since methods like Data Assimilation already model time varying processes, what is the potential benefit of the LSTM? This section should contain a more clear and comprehensive justification of the LSTM to motivate it over existing time series techniques.*

**A:** We will add a more clear justification for employing the Bi-LSTM in our manuscript. The main motivation of the usage of a purely data-driven method, in contrast to a data-assimilation methodology, is its simplicity in practice, as no climate model simulation is needed.

However, we do not totally agree with the reviewer that data-assimilation methods automatically take into account the serial correlation of the system. They do in the field of numerical weather prediction, for example, but off-line data assimilation in paleoclimate do not general consider serial correlations. A an usual set-up of the Kalman filter in paleoclimate data assimilation, the prior state PDF is sampled from a climate simulation and the posterior is updated using information from the proxy data. No information about the serial correlation flows into this updating step.

*7 Comparing Table 1 and 2, why is SD ratio replaced with RMSE, particularly since RMSE was not mentioned as a comparison metric in the beginning of Section 3. Also, RMSE needs to be defined or at spelled out once.*

**A:** We employ RMSE metric for quantifying the bias error between targets and reconstructions. We will define RMSE. The rationale for not using the RMSE metric for the spatially resolved reconstructions is that this metric is also determined by the variances of the target and the reconstructions, and this variance is spatially not uniform. This makes the interpretation of a spatial field of RMSE more difficult. As one of our main objectives is the correct replication of the amplitude of variations, we think that this information is better conveyed by separately showing the correlation and the ratio of variances. For singled valued spatial averages or indices, this problem is not present and we will then provide all three measures, correlation, ratio of variances and RMSE.

*8 Line 156 states that "We then perturb the ideal pseudo-proxies with Gaussian white noise ... with signal-to-noise ratio (SNR) values of 0.25, 0.5 and 1", but then later on line 306 it states "More realistic pseudo-proxies are those containing 80% Gaussian white noise contamination.", and it would seem that the 80% contamination is used in all of the experiments. How is this 80% number connected the previously stated SNR values?*

**A:** The noise level can be expressed using various definitions including SNR, variance of pure white noise (NVAR), and percentage of noise noise in the total variance variance (PNV) (Smerdon, 2012). Each of them can be readily translated: for example, 80% PNV corresponds to a SNR of 0.5. We will define PNV and use it uniformly through the manuscript to avoid confusion.

*9 On line 395 – "The Bi-LSTM is able to capture periods of extreme cooling better than the other two methods but strongly underestimates the recent warming trend." Is it possible the LSTM is just biased towards colder temperatures?*

**A**: We will test whether this property generally holds for different LSTM architectures or whether the LSTM is more sensitive than the other methods to the choice of the training period, since we used in the first version a generally colder training period. Another possibility is that the replication of extremes by this method is indeed not symmetrical. When analyzing the behavior of the methods and reconstructing extremes, we will also pay attention to any asymmetry in the reconstructed distribution functions. We will also investigate in a more detailed way whether the reconstruction skill is different for the multidecadal means or for extreme annual temperatures.

*10 The figures need to be referenced more heavily in the text. Statements such as In addition, cc maps show higher values over regions where more pseudoproxies are located." and The Bi-LSTM and PCR methods*

*exhibit relatively consistent patterns with similar SD ratios" seem to refer to the content of a plot. Without an explicit reference though its hard to follow.*

**A**: We will take care to reference the figures more frequently throughout the text.

*11 I think section 2.2.1. Construction of pseudo-proxies should be grouped in with the Data section 2.1, rather than the Methods section 2.2.*

**A:** This point can be considered a matter of taste. Actually, the data we used stem from climate simulations. The construction of pseudo-proxies is more a methodological issue, as pseudo-proxies can be constructed differently from the same underlying data. Our suggestions is to now include a Data and Methods section, separated in subsections.

**3. Technical corrections**

We will proofread the whole manuscript and correct grammatical and informal mistakes. We will make all changes considering the stated technical corrections.

Reference:

Najafabadi, M.M., Villanustre, F., Khoshgoftaar, T.M. et al. Deep learning applications and challenges in big data analytics. Journal of Big Data 2, 1 (2015). https://doi.org/10.1186/s40537-014-0007-7.

Riley, P. Three pitfalls to avoid in machine learning. Nature 572, 27–29 (2019).

Smerdon, J. E.: Climate models as a test bed for climate reconstruction methods: pseudoproxy experiments, Wiley Interdisciplinary Reviews, Clim. Change., 3, 63–77, https://doi.org/10.1002/wcc.149, 2012.

---

## Author Comment (AC2)

We would like to thank Referee 2 for their detailed and constructive comments.

**General comments**

C: *The application of Bi-LSTM is new, but the authors do not make a strong case for why this method should be applied. Of all the machine learning methods, why this one? Is Bi-LSTM particularly well suited for the CFR problem? Is the non-linear nature of the method or its incorporation of serial correlation important for the problem? Without strong arguments for why these characteristics are useful, the application of Bi-LSTM has the feeling of just being the method that the authors had sitting on the shelf. This should be remedied.*

A: We agree that there are several possible methods that could have been tested from the ML methodologies. From the available methods that we could have used for CFR, recurrent neural networks (RNNs) are potentially a suitable candidate with the objective of better reconstructing the true climate variability, because they can learn the serial correlation. Our underlying assumption, to be tested in this study, is that this property can improve the reconstruction variance. In general, RNNs learn only the short-term serial correlation (Bengio, et al. 1994). Bi-LSTM is a special type of RNN of which it has been demonstrated that it is capable of learning and capturing long-term dependencies from a sequential dataset (Hochreiter & Schmidhuber, 1997). The Bi-LSTM combines two independent LSTMs together, which allows the network to incorporate both backward and forward information for the sequential time series at every time step. Our working hypothesis is, that a more sophisticated type of RNN could better replicate the past variability, and perhaps even more so for extreme values. We would like to test whether this property of the Bi-LSTM is useful for paleo climate research in the future based on our experiments. In the revised version, we will add a more clear justification for why we have chosen this particular method for testing.

C: *Another general concern is that the manuscript is largely just an application of the methods and a description of the results. There is little insight into \*why\* the methods might behave the way that they do. Does the Bi-LSTM perform similar or worse to the PCR and CCA methods simply because the problem isn't strongly non-linear? Does Bi-LSTM capture the cold extremes \*because\* it is non-linear? If that is the case, why does it better capture the cold extremes and not the warm extremes? These and other questions are simply not taken up and the descriptive presentation of the results is not commensurate with the state of the science in terms of how the results are interpreted to understand how and why methods perform the way that they do. Consider some of the recent work in paleo data assimilation, in which the motivation is to incorporate new information in the form of climate model constraints, or in the Yun et al. (2021, https://doi.org/10.5194/cp-17-2583-2021) paper in which the authors seek to diagnose why various methods perform the way that they do.*

A: We thank the reviewer for the Yun et al (2021) reference. We think the for us most relevant aspect in Yun et al. study is the comparison of the distribution of reconstruction variance over the leading EOFs. In this regard, we need to consider that the PCR method is a direct reconstructions of the EOFs, and that the CCA method uses a PCA decomposition as one of its steps. Therefore, the Yun et al. evaluation is for those two linear methods not very informative, as they almost automatically produce reconstructions with the 'correct' EOF patterns - the corresponding explained variances may be informative though. The Yun et al. perspective is more informative for the Bi-LSTM method. We will therefore explore the EOFs and the variance spectrum in the reconstruction produced by this method in more detail.

We will also discuss our results in the context of data-assimilation methods (also in response to comments of Referee 1), considering their relative advantages and disadvantages, but of course a full general comparison deserves a whole study on its own.

Regarding the similar or worse performance of the LSTM compared to PCA/CCA: Our underlying assumption was that the relationship between proxies and climate is inherently non-linear and therefore a non-linear method should be able to capture the variability better. Therefore, the fact that the Bi-LSTM does not perform better than PCR/CCA can have two possible reasons: (1) the problem is simply not non-linear enough (as suggested by the reviewer) or (2) the Bi-LSTM architecture is not yet optimal. We will test this by further exploring the LSTM sensitivity to changes in architecture and parameters. If the LSTM results appear to be rather insensitive to changes in network architecture etc, we would conclude that indeed the problem at hand may not be non-linear enough to benefit from the non-linear properties of the Bi-LSTM.

In response to both reviewers, we will also change the calibration period to a more realistic interval in the 20$^{th}$ century (see also response to specific comment to l.165). Testing the sensitivity of the Bi-LSTM to the training period will also help us in understanding whether the capability of the Bi-LSTM to better reproduce **cold** events simply reflects the relatively colder training period, or whether there is a more systematic bias towards cold events (which we then wold need to explore further).

**Specific comments**

Ln 7: *There are many places in the manuscript that use "summer season temperature." Summer season is redundant and can be changed to simply summer temperature.*

**A:** We will change "summer season temperature" to summer temperature

Ln 22: *This is a bit of a strange list of references for this general statement. I suggest the authors just list the many review articles on CE climate over the last decade or more: Mann and Jones (2003); Jones et al. (2009); Frank et al. (2010); Christiansen and Ljungqvist (2012); Smerdon and Pollack (2016); Christiansen and Ljungqvist (2017)*

**A:** We will change the references and list more review articles on CE climate over the last decade

Ln 24: *Again, this is a strange list of references for the sentence it is supporting. More appropriate references are: Hegerl et al. (2006, 2007); Schurer et al. (2013, 2014); Anchukaitis et al. (2012, 2017); Tejedor et al. (2021 PNAS and PP)*

**A:** We will make changes correspondingly

Ln 26: *"hinders to capture" is not grammatically correct.*

**A:** We will correct this grammatical mistake.

Ln 28: *"ice cores), etc." is incorrect structure.*

**A:** We will correct this structure.

Ln 41-40: *"Many scientific studies that employ pseudo-proxies and real proxies have focused on global and hemisphere climate field or climate index reconstructions..." What else is there? This is basically*

*everything unless the authors are thinking about recons of dynamic indices or want to point out that the majority have focused on specifically temperature recons (with the exceptions of the data assimilation methods that have tested multiple variable recons in pseudoproxy studies).*

A: We were referring to regional reconstructions that span smaller regions than the globe or one hemisphere, e.g. the North Atlantic. The PAGES2k reconstructions do consider continental scales, but essentially they include just one (or two in some cases) continental-scale reconstructions.

Ln 51: *Data assimilation isn't mentioned at all in the Introduction, which overlooks a rapidly expanding area of CFR research and production in the field right now. It is relevant here inasmuch as the method does not assume temporally stable relationships between proxies and the targeted climate variables.*

**A:** Following the advice of both reviewers, we will include a discussion of data assimilation methodologies and summarize the difference to our approach. The family of data-assimilation methods is a different approach and is based on a rather different data set, as it also requires data from climate simulations. In doing so, it is not constrained, in principle, by the assumption that the spatial covariance is stationary over time, but on the other hand, some data-assimilation methods based on Kalman Filters require some sort of regularization of the error-covariance matrices. They also might be computationally much more demanding that purely data-driven methods. Also importantly, the use of information from climate models precludes that the resulting reconstructions might be used to test the simulations themselves, and so the resulting reconstructions lose part of their original justification.

We will include a brief discussion about the pros and cons of purely data-driven and data-assimilation methods, and the skill of our reconstructions will be also discussed in the light of the data-assimilation reconstructions. but the objective of this paper is indeed not this type of comparison, which would require its own dedicated study.

Ln 53: *Coats et al. (2013, doi:10.1002/grl.50938) and Yun et al. (2021) specifically take up the stationarity assumption.*

**A:** We will correct the sentence and refer these two articles

Ln 71: *Decadally filtered after the forced global warming signal has been removed (usually via detrending).*

A: We think that the question of detrending the SST data to define the AMV index in the paleoclimate context is not as clear as for the recent climate (20th century). The purpose of detrending is to eliminate the impact of anthropogenic climate change and in theory retain only the part of variations driven by natural variability. The role of aerosols remains contested, as this cannot be easily removed by detrending alone. The reviewer is surely aware of the ongoing discussions on the origin of the AMV variability (either externally forced or internally generated, e.g Booth et al., 2012 ; Clement et al, 2015) . In the paleoclimate context, these issues are not directly relevant. Anthropogenic climate change is absent or much weaker, and thus all AMV variations are naturally generated. If one would like to focus on the internally generated variations only, a model to subtract the impact of the forcing would be needed, and this is prone to errors when the external forcing is not exactly known (Mann et al., 2022). However, for the present study this distinction is not really important, as the objectives is to reconstruct the AMV variations independently of their origins. Another practical problem would be the definition of the detrending period in the past millennium. A linear trend over the whole period is not really justified. Therefore, we think it is reasonable to stick to our initial definition of the AMV, which has also been adopted in other paleoclimate studies, e.g. Wang et al., 2017)

Ln 91: *It is not clear what is being combined here. It was stated above that they use the PAGES2k network combined with a tree-ring network from St. George. Here they say they are combining mollusk shell records with PAGES2k and Luterbacher et al. The inclusion of the mollusk shell records is a bit random, as I am not sure they have been included in a large-scale CFR to this point. In a synthetic experiment like this, it seems a bit ad hoc to create a sampling based on a theoretical combination of proxies that, to my knowledge, has never been adopted before (I am not aware of a large-scale application of the St. George assessment, unless the authors are using that reference to refer to a large-scale sampling from the ITRDB). Just using the PAGE2k sampling seems sufficient and straightforward here.*

**A:** We do not totally agree with this comment. One objective of our study is the reconstruction at smaller scale than hemispheric, e.g. at European continental scales. The PAGES2k network for Europe is constrained to land proxies. On the other hand, the OCEAN2k Pages reconstructions include marine proxies. We think that an interesting question is to explore the reconstruction skill when both sort of proxies are combined. The reviewer is right in that the mollusk shell records have not yet been included in larger compilation of proxies, but we find no compelling reason not to do it now.

Thus the objective is explore the combination of suitable terrestrial and marine proxies for small-scale North Atlantic and European land surface temperature reconstructions, while for the large-scale Northern Hemisphere surface temperature reconstruction we use a subset of the PAGES2k network. This subset is defined following the assessment by St. George, S (2014). He analyzed all tree ring data from ITRDB, and foundthat some tree ring proxies in the mid-latitudes of America and Eurasia are less suitable for temperature reconstructions because they show a more positive correlationwith summer precipitation instead of summer temperature (Figure 4 of St. George, S. 2014). We therefore think that the comprehensive criteria by St. Georg can help filter the proxy data from the PAGES2k network.

However, we would like to point out that a particular choice of proxy network cannot be critical to evaluate reconstructions methods, as these networks can be updated and re-evaluated. The main methodological conclusions should be broadly independent of the pseudo-proxy network used. To test this hypothesis, we offer to perform a test reconstruction based on the original PAGES2k network and compare it with the results of our filtered network.

Ln 95: *The choice of climate model has been definitively shown to impact the pseudoproxy results: Smerdon et al. (2011, 2015) and Parsons et al. (2021, https://doi.org/10.1029/2020EA001467)*

**A:** We will correct this sentence and refer the articles mentioned here.

Ln 132: *Which ensemble member? Also from not form.*

**A:** We will add the ensemble member and correct this grammatical mistake

Ln 134: *The CCSM4 model is presented as if it is distinct from the CESM, when in fact they share a very close lineage. This should be mentioned and does not make what the authors have done to be three truly independent models because of the close lineage between CCSM4 and CESM.*

**A:** We agree with this comment and we will limit our analysis to the two clearly different climate models MPI-ESM and CESM in the revised version. This should also reduce the descriptive nature of the manuscript and give more space for the exploration of the reasons for the respective method's performance.

Ln 150: *The grid cell that contains the proxy location is probably more accurate.*

**A:** We will correct the expression correspondingly.

Ln 156: *It is useful to point out that white noise is not realistic and that there have been attempts to use other noise colors or noise simulated by proxy system models. Noise colors were investigated in the seminal von Storch et al. (2004) paper, Wang et al. (2014, https://doi.org/10.5194/cp-10-1-2014) investigated various noise structures, and Evans et al. (2014, https://doi.org/10.1002/2014GL062063) investigated pseudoproxy experiments with noise from proxy system models.*

**A:** We will include experiments with red-noise contamination. In general, the results with red-noise pseudoproxies is to further reduce the reconstructed variance, but we will test whether this also applies to the LSTM network.

Ln 165: *I strongly disagree with what the authors have done to split up the calibration and validation intervals. They use a much longer training interval than would ever be possible in the real world and they calibrate outside of the 20th century when the strong trend therein may have important impacts on their methodological performance. Given the descriptive nature of this study, it is weakened even more if the conditions under which the methods are tested are far outside of what is possible with real data. I strongly encourage the authors to complete the study over a more realistic calibration interval length and in the 20th century. Absent these more realistic constraints, the skill measures the authors provide are probably inflated and impossible to interpret for more realistic frameworks.*

**A:** In the revised version, we will test the CFR methods with a more realistic calibration interval length from the 20th century and update the skill metrics and discussion accordingly

Equation 1: *The PCR and CCA formalism is inexplicably written in series form. Why not use the much more traditional formulation using matrix notation? The relationship between PCR and CCA is also more evident using matrix notation, in which PCR is simply a special form of CCA, i.e. it does not reduce the rank of the cross correlation matrix. This relationship should also be noted.*

**A:** We agree with the reviewer that a matrix notation is more compact and probably elegant. However, we need to also consider that some readers of this manuscript may be more familiar with the series form. We have in mind research groups working on climate reconstructions, familiar with calibration of dendrochronological series and that may be not that familiar with the matrix notation, but who nevertheless may be interested in new applications of CFR. We surmise that the best notation in each case is a matter of personal taste, and that the important point is that all mathematical steps remain clear.

Ln 192: *Residual term with what assumed properties?*

**A:** The residual could be an unobserved random variable that adds noise to the linear relationship between the dependent variable (PC) and the targeted regressors (proxy or pseudoproxy) and includes all effects on the targeted regressors not related to the dependent variable (Christiansen, 2011). We agree with the reviewer that the statistical properties of the assumed noise can make a difference to the reconstruction skill. We will perform experiments with red noise in addition to the white noise already included

Paragraph starting on Ln 279: *This paragraph is full of undefined jargon that is not cited. It is meaningless for the uninitiated. Please correct.*

**A:** We will revise this paragraph starting on line 279 to better clarify the Bi-LSTM network, adding citations (Knerr, et, al. 1990; Yu, et, al. 2019) and removing the amount of jargon as much as possible.

Ln 299: *This was first noted in Smerdon et al. (2010, 2011).*

**A:** We will refer Smerdon et al. (2010, 2011).

Ln 310: *"reduction in skill" as opposed the vague use of degradation here?*

**A:** We will correct degradation to reduction in skill.

Ln 320: *In the spirit of my general comments, one curiosity is why CCA does not perform better than PCR with regard to the cc metric. CCA is designed to optimize the correlation, which is why it can sometimes yield larger variance losses. It is therefore curious why it doesn't universally beat out PCR in the cc metric.*

**A:** CCA indeed maximizes the correlation that can be attained with a change of variables, i.e. with a linear combination of the grid-cell series in each of the two fields. In an admittedly artificial example, the canonical correlation can be very high and yet the reconstruction skill in general can remain low. If one grid cell in each field are very highly correlated to one another (and assuming here no PCA pre-filtering), CCA will pick those two cells as the first CCA pair (i.e., a pattern in each field with very high loadings on those cells). The rest of the cells will not contribute to the CCA pattern. The reconstruction skill will therefore be very low in all those cells, despite the CCA being very high. In general, the reconstruction skill will be a monotonic function of the canonical correlation coefficient and the variance explained by the canonical predictand pattern. If the latter is low, the reconstruction skill will be low in large areas of the predictand field.

We will introduce a short discussion in the manuscript to guide the reader.

Ln 327: *The variance losses have a relatively straightforward interpretation for the traditional regression approaches. When analyzing the mean results, the variance losses reflect loss of signal (reflected in the mean) and increases in the variance associated with the error term. It would be useful to know if the machine learning method can be interpreted in the same way, or if there is an alternative way to think about variance losses for that method.*

**A:** We believe that in the case of machine learning algorithms, but the interpretation will be more difficult. We also point out here that, even in the reviewer's interpretation, simple algorithms can yield contrasting results. For instance, even assuming that noise is present in the predictand only, direct linear regression and inverse linear regression lead to under and over estimation of the reconstructed variance. In the case of neural network, it is very common that they also lead to underestimation of the variance (noted already long time ago, Zorita et al., 1999 https://doi.org/10.1175/1520-0442(1999)012%3C2474:TAMAAS%3E2.0.CO;2), but this can depend on the structure of the network and specifically on the form of the prescribed neuron activation function.

We will discuss this point in the manuscript, but a general answer would be, in our view, rather complex.

Ln 390: *Why should complexity translate to improved skill? I am aware of no grand postulate that makes this case.*

**A:** The reviewer is correct that there is no general principle linking complexity and skill, but it is reasonable to assume that a more complex model might be able to better capture more complex relationships. For instance, a linear model cannot capture non-linear links outside a narrow range. Artificial neural network is a subset of machine learning method could usually be understood as a universal approximator which can map and approximate any kind of linear or nonlinear functions by selecting a suitable set of connecting weights and transfer functions in principle (Hornik et al., 1989).

Thus, it is reasonable to assume that a better performance might be achieved, but indeed, this not generally guaranteed. We will reformulate this sentence.

Ln 402: *The relationship (or lack thereof) between the skill of the mean indices and spatial skill was first discussed in Smerdon et al. (2010, 2011) and further highlighted in Smerdon's 2012 pseudoproxy review.*

**A:** We will make corrections and refer the articles mentioned here.

Ln 418: *This is vague. What about alternative methods might be useful in the context of the CFR problem? There are lots of methods out there. What direction can the authors provide, based on the work they have done, that might represent useful characteristics in other machine-learning methods to try in the context of this problem?*

**A:** We will present alternative directions and methods that might be useful in paleo CFR experiments based on more realistic calibration and validation time period as the reviewer suggested. Our first implementation of the more complex Bi-LSTM does not show superiority in CFRs compared to traditional CFR methods, so we would like to draw an assumption that more complicated architecture might not be helpful for CFRs at least based on our specific experiment results and the employed architecture of Bi-LSTM. However, we would suggest an Echo State Network (ESN) for paleo climate research. (Lukosevicius, M. & Jaeger, H. 2009; Nadiga, 2020). Both ESN and LSTM belong to RNN, yet ESN is much simpler than LSTM (Lukosevicius, M. & Jaeger, H. 2009), and also has outperformed the RNN methods in other applications (Chattopadhyay et al., 2019, Nadiga, B. 2020). We thus encourage testing ESN in different paleo climate research directions.

Ln 451: *CCA is a classic linear-based CFR method. This structure is awkward.*

**A:** We will correct this structure.

Ln 460: *"Reservoir Computing methods-Echo State Network" is screaming for a reference so that the rest of us can figure out what it is.*

**A:** We will add references about ESN: e.g., Lukosevicius, M. & Jaeger, H. 2009 and Nadiga, 2020

Figures 6 and 7: *Much of the text in this figure would only be legible by Ant Man. I strongly suggest increasing the size of the legend, fonts, and axis labels.*

**A:** We will increase the size of legend, fonts and axis labels .

Figures 8 and 9: *I find these figures very hard to read. Why include the bar plots for the data bins? It would be much clearer to simply show the estimated PDFs, which characterize the behavior well enough.*

A: The reason for including the bin bars is to highlight the differences in the frequency of extremes. The smoothed PDFs can provide a picture that is not totally accurate. Our suggestion is to produce pictures that show the smoothed PDFs and, additionally, a few bin bars for extreme events

**References:**

Bengio, Y., Simard, P. & Frasconi, P. Learning long-term dependencies with gradient descent is difficult. IEEE Trans. Neural Networks 5, 157–166 (1994).

Hochreiter, S. & Schmidhuber, J. Long short-term memory. Neural Comput. 9, 1735–1780 (1997). This paper introduced LSTM recurrent networks, which have become a crucial ingredient in recent advances with recurrent networks because they are good at learning long-range dependencies.

Hornik, K, Stinchcombe, M, White, H. Multilayer feedforward networks are universal approximators. Neural Networks, 2 (1989), pp. 359-366.

Rasp, S. & Lerch, S. Neural networks for postprocessing ensemble weather forecasts. Mon. Weather Rev. 146, 3885–3900 (2018).

Castelvecchi, D. Can we open the black box of AI? Nature 538, 20–23 (2016).

Toms, B. A., Barnes, E. A., & Hurrell, J. W. (2021). Assessing decadal predictability in an Earth-system model using explainable neural networks. Geophysical Research Letters, 48, e2021GL093842. https://doi.org/10.1029/2021GL093842

Molnar, C. Interpretable Machine Learning—A Guide for Making Black Box Models Explainable. 2019. Available online: https://christophm.github.io/interpretable-ml-book/

Christiansen, B.: Reconstructing the NH mean temperature: can underestimation of trends and variability be avoided?, J. Clim., 24, 674–692, 2011.

Knerr, S., Personnaz, L and Dreyfus, G., "Single-layer learning revisited: A stepwise procedure for building and training a neural network" in Neurocomputing: Algorithms Architectures and Applications, Berlin, Germany:Springer, pp. 41-50, 1990, [online] Available: http://dx.doi.org/10.1007/978-3-642-76153-9-5.

Yu, Y., Si, X., Hu, C. & Zhang, J. A review of recurrent neural networks: LSTM cells and network architectures. Neural Comput. 31, 1–36 (2019).

Lukoševičius, M. & Jaeger, H. Reservoir computing approaches to recurrent neural network training. Computer Science Review 3, 127–149 (2009).

Nadiga, B.: Reservoir Computing as a Tool for Climate Predictability Studies, J. Adv. Model. Earth Sy., e2020MS002290, https://doi.org/10.1029/2020MS002290, 2020.

Chattopadhyay, A., Hassanzadeh, P., and Subramanian, D.: Data-driven predictions of a multiscale Lorenz 96 chaotic system using machine-learning methods: reservoir computing, artificial neural network, and long short-term memory network, Nonlin. Processes Geophys., 27, 373–389, 2020.

---

## Author Response (AR1)

We would like to thank Referee 1 for their detailed and constructive comments. In the following to explain how we have modified the manuscript to address their suggestions.

We have considerably revised the manuscript. The revisions can be summarized as follows:

1) We have now considerably expanded the range configurations of the Bi-LSTM method

2) We have changed the calibration period of the pseudo-proxy experiements to roughly match the observational period.

3) We include now experiments with red-noise pseudo-proxies

4) We include a new section to analyze the spatial co-variability modes of the reconstructed fields, as in Yun et al. (2021)

**The original reviewer's suggestions are written in bold font**, and our responses with normal font.

**1. **General comment**

In response to their general comment, we have now deepened our exploration of the LSTM sensitivity to different architectures and tunable parameters. In the revised version, we have evaluated the LSTM methodology using a range of architectures. All versions of the Bi-LSTM method behave similarly in general, with only unsystematic variations. We conclude that the underlying LSTM structure does not provide the benefits we had expected, although the method itself provides results with comparable quality as the traditional methods.

**2. Specific comments**

**1. I'm still unclear as to whether this article is proposing Bi-LSTMs as a viable alternative to traditional statistical methods or whether its trying to show that they don't work well enough. If this article is intended to propose using Bi-LSTMs (or to refute their use) then this stance needs to be made more clear up front and in the results. Right now the article presents the results as a neutral comparison of methods", but this makes it difficult to reach a conclusion about the proposed method.**

A: The revised version explores more broadly the Bi-LSTM method, varying the number of layers, size of layers, cost function, and other tunable parameters. The results achieved so far display no systematic variations of the results achieved with the Bi-LSTM method, so that our conclusion is that a reasonable architecture will already provide reasonable results. We also conclude, however, that we were not able to achieve a clear improvement by varying the Bi-LSTM network. This might be due to the complexity of tuning this method to achieve a particular target, or it may be due to our too optimistic initial assumption that capturing the serial correlation present in the time series could provide a better replication of the amplitude of past variations.

**2 Follow up to Q1-If the article is proposing (or refuting) Bi-LSTMs then it needs a more comprehensive evaluation of the Bi-LSTM model. The results in Appendix B are a good start, but it would have been nice to see how varying the depth of the network, using dropout, weight decay, or other regularization techniques, or varying the learning rates (or using a scheduler), number of epochs, and other aspects of the training procedure such as the loss function effect generalization. From this, a reader could draw broader conclusions about the effectiveness of Bi-LSTM models rather than the effectiveness of the single model presented.**

**A**: See our previous response. We have explored a range of architectures, different network depths, introducing dropout layers, using different learning rates, and employing different loss functions to provide a more comprehensive evaluation of the Bi-LSTM performance and effectiveness. We include most of these results in an appendix, and retain in the main manuscript only those results/configurations that help support the main conclusions on the Bi-LSTM method.

**3 The statement "The reconstruction of mean temperature series could provide a general assessment of the skill to reconstruct extreme temperature phases" needs either a citation or experimental results**

**in Section 3. I think extremal behavior could be quite different than behavior near the mean? It would be interesting to see if the Bi-LSTM can model quantiles of the distribution better than PCR or CCA.**

A: As we mentioned in our initial reaction to the review, that this sentence was meant to justify the consideration of the Bi-LSTM method in the first place. This justification is now more explicit in the introduction This method, in contrast to the usual set-up with the traditional PCR and CCA methods, naturally incorporates the serial correlations of the inputs. The working hypothesis is that this could improve the reconstruction of extremes, or at least provide a different reconstruction skill than for the mean values. Our objective was to test this potential difference.

**4 What is the rationale for training on 850-1425 and then testing on the later period of 1426-2000 (where did 2000-2005 go?), rather than the reverse as in Steiger et al. (2014)? I think that in a paleoclimate experiment we would be more interested in the performance of our method in the relative past, rather than the relative future. Would the performance of different methods change with the temporal order of training and testing?**

A: We have now replaced the calibration period with the 20 century, and use the rest of the past last millennium time period as validation (also in response to a comment by Reviewer 2).

We have included a note of caution in the introduction on using this short calibration period, as it is by necessity done in real reconstructions. Some methods such as PCR can be comprised by overfitting if a number of predictors, due to a large proxy network, is comparable to the number of samples in the calibration period.

**5 In practice, the Bi-LSTM would need to be trained on real proxies and real observations, which would limit the training period to 1850 onwards. Will this be enough observations, over a long enough time horizon, to train an LSTM model? Comparisons between the various methods under this limited data setting would be helpful. Also, how will you account for the significant covariate shift between the post-industrial land pre-industrial periods?**

A: As per our previous response, we now use a realistic calibration period to test the methods in more realistic circumstances.

**6 Lines 225-235 seem to motivate including temporal correlation in a model more generally, rather than LSTMs specially. Since methods like Data Assimilation already model time varying processes, what is the potential benefit of the LSTM? This section should contain a more clear and comprehensive justification of the LSTM to motivate it over existing time series techniques.**

A: We have added a more clear justification, actually an expectation, for employing the Bi-LSTM in our manuscript. This has been shortly explained in the previous responses. Regarding additional justifications w.r.t. to data assimilation methods, we now explain in the manuscript two. One is that data assimilation methods require data from climate simulations, This makes the method itself more cumbersome. A second justification is that this use of data from climate simulations precludes a clean assessment of reconstructions vis-a-vis simulations, which is one of the most important reasons for climate reconstructions in the first place.

**7 Comparing Table 1 and 2, why is SD ratio replaced with RMSE, particularly since RMSE was not mentioned as a comparison metric in the beginning of Section 3. Also, RMSE needs to be defined or at spelled out once.**

A: This was indeed somewhat confusing. The manuscript is now focused on the SD ratio for spatially resolved reconstructions and in addition, and only for the index reconstructions is the RMSE additionally considered. As we explained in our first reaction to the reviewer's comments, this is because the RSME is not as informative when the variance varies across space.

**8 Line 156 states that "We then perturb the ideal pseudo-proxies with Gaussian white noise ... with signal-to-noise ratio (SNR) values of 0.25, 0.5 and 1", but then later on line 306 it states "More realistic pseudo-proxies are those containing 80% Gaussian white noise contamination.", and it would seem that the 80% contamination is used in all of the experiments. How is this 80% number connected the previously stated SNR values?**

**A:** The noise level can be expressed using various definitions including SNR, variance of pure white noise (NVAR), and percentage of noise noise in the total variance variance (PNV) (Smerdon, 2012). Each of them can be readily translated: for example, 80% PNV corresponds to a SNR of 0.5. We now define PNV and use it uniformly through the manuscript to avoid confusion.

**9 On line 395 – "The Bi-LSTM is able to capture periods of extreme cooling better than the other two methods but strongly underestimates the recent warming trend." Is it possible the LSTM is just biased towards colder temperatures?**

**A**: As we changed the calibration period to the observational period to match more realistically the conditions of real reconstructions, this property is not as obvious as when using a cold calibration period. It would be indeed interesting to assess the implications of a longer colder calibration period, but both reviewers expressed a strong recommendation to stick to the observational period.

**10 The figures need to be referenced more heavily in the text. Statements such as In addition, cc maps show higher values over regions where more pseudoproxies are located." and The Bi-LSTM and PCR methods exhibit relatively consistent patterns with similar SD ratios" seem to refer to the content of a plot. Without an explicit reference though its hard to follow.**

**A**: We have now cite  the figures more frequently throughout the text.

**11 I think section 2.2.1. Construction of pseudo-proxies should be grouped in with the Data section 2.1, rather than the Methods section 2.2.**

**A:** This point can be considered a matter of taste. Actually, the data we used stem from climate simulations. The construction of pseudo-proxies is more a methodological issue, as pseudo-proxies can be constructed differently from the same underlying data. The new version of the manuscript  now includes a Data and Methods section, separated in subsections.

**3. Technical corrections**

We have proofread the whole manuscript and expect  the number of grammatical errors to be much smaller

We would like to thank Referee 2 for their detailed and constructive comments. In the following to explain how we have modified the manuscript to address their suggestions.

We have considerably revised the manuscript. The revisions can be summarized as follows:

1) We have now considerably expanded the range configurations of the Bi-LSTM method

2) We have changed the calibration period of the pseudo-proxy experiements to roughly match the observational period.

3) We include now experiments with red-noise pseudo-proxies

4) We include a new section to analyze the spatial co-variability modes of the reconstructed fields, as in Yun et al. (2021)

**The original reviewer's suggestions are written in bold font** and our responses with normal font.

**General comments**

**C: The application of Bi-LSTM is new, but the authors do not make a strong case for why this method should be applied. Of all the machine learning methods, why this one? Is Bi-LSTM particularly well suited for the CFR problem? Is the non-linear nature of the method or its incorporation of serial correlation important for the problem? Without strong arguments for why these characteristics are useful, the application of Bi-LSTM has the feeling of just being the method that the authors had sitting on the shelf. This should be remedied.**

**A:** We agree that there are several possible methods that could have been tested from the ML methodologies. From the available methods that we could have used for CFR, recurrent neural networks (RNNs) are potentially a suitable candidate with the objective of better reconstructing the true climate variability, because they can capture serial correlation. Our underlying assumption, to be tested in this study, is that this property can improve the reconstruction variance. In general, RNNs learn only the short-term serial correlation (Bengio, et al. 1994). Bi-LSTM is a special type of RNN of which it has been demonstrated that it is capable of learning and capturing long-term dependencies from a sequential dataset (Hochreiter & Schmidhuber, 1997). The Bi-LSTM combines two independent LSTMs together, which allows the network to incorporate both backward and forward information for the sequential time series at every time step. Our working hypothesis is, that a more sophisticated type of RNN could better replicate the past variability, and perhaps even more so for extreme values. We would like to test whether this property of the Bi-LSTM is useful for paleo climate research in the future based on our experiments.

Therefore, the choice of the Bi-LSTM method is not as specific as the first version of the manuscript perhaps conveyed. This method happens to be one that, in principle, is one of the best suited to capture the temporal structure of the time series. We have expanded this justification in the introduction.

**C: Another general concern is that the manuscript is largely just an application of the methods and a description of the results. There is little insight into \*why\* the methods might behave the way that they do. Does the Bi-LSTM perform similar or worse to the PCR and CCA methods simply because the problem isn't strongly non-linear? Does Bi-LSTM capture the cold extremes \*because\* it is non-linear? If that is the case, why does it better capture the cold extremes and not the warm extremes? These and other questions are simply not taken up and the descriptive presentation of the results is not commensurate with the state of the science in terms of how the results are interpreted to understand how and why methods perform the way that they do. Consider some of the recent work in paleo data assimilation, in which the motivation is to incorporate new information in the form of**

**climate model constraints, or in the Yun et al. (2021, https://doi.org/10.5194/cp-17-2583-2021) paper in which the authors seek to diagnose why various methods perform the way that they do.**

**A:** The manuscript now includes a completely new section on the compariosn of the patterns of spatial covariability, as in Yun et al. (2021). We thank the reviewer for the Yun et al (2021) reference. We think the for us most relevant aspect in Yun et al. study is the comparison of the distribution of reconstruction variance over the leading EOFs for the Bi-LSTM methods, as the other traditional linear methods have an in built advanatge in this regard (the regression is conducted already in the EOF space). .Therefore, the Yun et al. evaluation is for those two linear methods not very informative, as they almost automatically produce reconstructions with the 'correct' EOF patterns - the corresponding explained variances may be informative though. The Yun et al. perspective is more informative for the Bi-LSTM method.

We also include a brief discussion of the conclusions that can be derived from the comparison of reconstructed EOFs patters.

We now briefly discuss data-assimilation methods in the introduction (also in response to comments of Referee 1), considering their relative advantages and disadvantages, but of course a full general comparison deserves a whole study on its own.

Regarding the similar or worse performance of the LSTM compared to PCA/CCA: Our underlying assumption was that the relationship between proxies and climate is inherently non-linear and therefore a non-linear method should be able to capture the variability better. Therefore, the fact that the Bi-LSTM does not perform better than PCR/CCA can have two possible reasons: (1) the problem is simply not non-linear enough (as suggested by the reviewer) or (2) the Bi-LSTM architecture is not yet optimal. We have tested this second hypothesis by further exploring the LSTM sensitivity to changes in architecture and parameters. The LSTM results appear to be rather insensitive to changes in network architecture etc, so that we may conclude that this second hypothesis cannot explain the behaviour of the chosen Bi-LSTM configuration. A more likely explanation is that the reconstruction problem analyzed in this study is not non-linear enough, in otehr words, the pseudo-proxies are constructed in still a rather simple way. However, it is difficut to assess the possible non-itineraries in the real proxies. For this purpose w would need to perform a detailed analysis of real proxy series

In response to both reviewers, we have also changed the calibration period to a more realistic interval in the 20$^{th}$ century (see also response to specific comment to l.165).

**Specific comments**
**Ln 7: There are many places in the manuscript that use "summer season temperature." Summer season is redundant and can be changed to simply summer temperature.**
**A:** We have changed "summer season temperature" to summer temperature

**Ln 22: This is a bit of a strange list of references for this general statement. I suggest the authors just list the many review articles on CE climate over the last decade or more: Mann and Jones (2003); Jones et al. (2009); Frank et al. (2010); Christiansen and Ljungqvist (2012); Smerdon and Pollack (2016); Christiansen and Ljungqvist (2017)**
**A:** We have changed the references and list more review articles on CE climate over the last decade

**Ln 24: Again, this is a strange list of references for the sentence it is supporting. More appropriate references are: Hegerl et al. (2006, 2007); Schurer et al. (2013, 2014); Anchukaitis et al. (2012, 2017); Tejedor et al. (2021 PNAS and PP)**
**A:** We have include those changes

**Ln 26: "hinders to capture" is not grammatically correct.**
**A:** Noted and corrected

**Ln 28: "ice cores), etc." is incorrect structure.**
**A:** noted and corrected

**Ln 41-40: "Many scientific studies that employ pseudo-proxies and real proxies have focused on global and hemisphere climate field or climate index reconstructions..." What else is there? This is basically everything unless the authors are thinking about recons of dynamic indices or want to point out that the majority have focused on specifically temperature recons (with the exceptions of the data assimilation methods that have tested multiple variable recons in pseudoproxy studies).**
A: We were referring to regional reconstructions that span smaller regions than the globe or one hemisphere, e.g. the North Atlantic. The PAGES2k reconstructions do consider continental scales, but essentially they include just one (or two in some cases) continental-scale reconstructions.

**Ln 51: Data assimilation isn't mentioned at all in the Introduction, which overlooks a rapidly expanding area of CFR research and production in the field right now. It is relevant here inasmuch as the method does not assume temporally stable relationships between proxies and the targeted climate variables.**
**A:** Following the advice of both reviewers, we have now included a brief introduction of data assimilation methodologies and summarize the difference to our approach. Specifically, we now mention the advantages of those methods (powerful combination of all information available) and drawbacks (no critical evaluation of reconstructions and simulations possible any more). Regarding the isse of under or over estimation of variance,no systematic evaluation of data-assimilation methods is possible, as this property depends on the climate simulation used and on the stated uncertainties in the proxy-base reconstructions

**Ln 53: Coats et al. (2013, doi:10.1002/grl.50938) and Yun et al. (2021) specifically take up the stationarity assumption.**
**A:** Noted and corrected

**Ln 71: Decadally filtered after the forced global warming signal has been removed (usually via detrending).**
A: We think that the question of detrending the SST data to define the AMV index in the paleoclimate context is not as clear as for the recent climate (20th century). The purpose of detrending is to eliminate the impact of anthropogenic climate change and in theory retain only the part of variations driven by natural variability. The role of aerosols remains contested, as this cannot be easily removed by detrending alone. The reviewer is surely aware of the ongoing discussions on the origin of the AMV variability (either externally forced or internally generated, e.g Booth et al., 2012; Clement et al, 2015) . In the paleoclimate context, these issues are not directly relevant. Anthropogenic climate change is absent or much weaker, and thus all AMV variations are naturally generated. If one would like to focus on the internally generated variations only, a model to subtract the impact of the forcing would be needed, and this is prone to errors when the external forcing is not exactly known (Mann et al., 2022). However, for the present study this

distinction is not really important, as the objectives is to reconstruct the AMV variations independently of their origins. Another practical problem would be the definition of the detrending period in the past millennium. A linear trend over the whole period is not really justified. Therefore, we think it is reasonable to stick to our initial definition of the AMV, which has also been adopted in other paleoclimate studies, e.g. Wang et al., 2017)

**Ln 91: It is not clear what is being combined here. It was stated above that they use the PAGES2k network combined with a tree-ring network from St. George. Here they say they are combining mollusk shell records with PAGES2k and Luterbacher et al. The inclusion of the mollusk shell records is a bit random, as I am not sure they have been included in a large-scale CFR to this point. In a synthetic experiment like this, it seems a bit ad hoc to create a sampling based on a theoretical combination of proxies that, to my knowledge, has never been adopted before (I am not aware of a large-scale application of the St. George assessment, unless the authors are using that reference to refer to a large-scale sampling from the ITRDB). Just using the PAGE2k sampling seems sufficient and straightforward here.**
**A:** We do not totally agree with this comment. One objective of our study is the reconstruction at smaller scale than hemispheric, e.g. at European continental scales. The PAGES2k network for Europe is constrained to land proxies. On the other hand, the OCEAN2k Pages reconstructions include marine proxies. We think that an interesting question is to explore the reconstruction skill when both sort of proxies are combined. The reviewer is right in that the mollusk shell records have not yet been included in larger compilation of proxies, but we find no compelling reason not to do it now.
Thus the objective is explore the combination of suitable terrestrial and marine proxies for small-scale North Atlantic and European land surface temperature reconstructions, while for the large-scale Northern Hemisphere surface temperature reconstruction we use a subset of the PAGES2k network. This subset is defined following the assessment by St. George, S (2014). He analyzed all tree ring data from ITRDB, and foundt hat some tree ring proxies in the mid-latitudes of America and Eurasia are less suitable for temperature reconstructions because they show a more positive correlationwith summer precipitation instead of summer temperature (Figure 4 of St. George, S. 2014). We therefore think that the comprehensive criteria by St. Georg can help filter the proxy data from the PAGES2k network.
However, we would like to point out that a particular choice of proxy network cannot be critical to evaluate reconstructions methods, as these networks can be updated and re-evaluated. The main methodological conclusions should be broadly independent of the pseudo-proxy network used. To test this hypothesis, we have included a test reconstruction based on the original PAGES2k network and compare it with the results of our filtered network.

**Ln 95: The choice of climate model has been definitively shown to impact the pseudoproxy results: Smerdon et al. (2011, 2015) and Parsons et al. (2021, https://doi.org/10.1029/2020EA001467)**
**A:** Noted and corrected

**Ln 132: Which ensemble member? Also from not form.**
**A:** Noted and corrected

**Ln 134: The CCSM4 model is presented as if it is distinct from the CESM, when in fact they share a very close lineage. This should be mentioned and does not make what the authors have done to be three truly independent models because of the close lineage between CCSM4 and CESM.**
**A:** We agree with this comment and we have limited our analysis to the two clearly different climate models MPI-ESM and CESM in the revised version.

**Ln 150: The grid cell that contains the proxy location is probably more accurate.**

**A:** Noted and corrected

**Ln 156: It is useful to point out that white noise is not realistic and that there have been attempts to use other noise colors or noise simulated by proxy system models. Noise colors were investigated in the seminal von Storch et al. (2004) paper, Wang et al. (2014, https://doi.org/10.5194/cp-10-1-2014) investigated various noise structures, and Evans et al. (2014, https://doi.org/10.1002/2014GL062063) investigated pseudoproxy experiments with noise from proxy system models.**
**A:** We have now included experiments with red-noise contamination. In general, the results with red-noise pseudoproxies is to further reduce the reconstructed variance,

**Ln 165: I strongly disagree with what the authors have done to split up the calibration and validation intervals. They use a much longer training interval than would ever be possible in the real world and they calibrate outside of the 20th century when the strong trend therein may have important impacts on their methodological performance. Given the descriptive nature of this study, it is weakened even more if the conditions under which the methods are tested are far outside of what is possible with real data. I strongly encourage the authors to complete the study over a more realistic calibration interval length and in the 20th century. Absent these more realistic constraints, the skill measures the authors provide are probably inflated and impossible to interpret for more realistic frameworks.**
**A:** In the revised version, we have modified the calibration period (observational period) to match the conditions for real reconstructions

**Equation 1: The PCR and CCA formalism is inexplicably written in series form. Why not use the much more traditional formulation using matrix notation? The relationship between PCR and CCA is also more evident using matrix notation, in which PCR is simply a special form of CCA, i.e. it does not reduce the rank of the cross correlation matrix. This relationship should also be noted.**
**A:** We agree with the reviewer that a matrix notation is more compact and probably elegant. However, we need to also consider that some readers of this manuscript may be more familiar with the series form. We have in mind research groups working on climate reconstructions, familiar with calibration of dendrochronological series and that may be not that familiar with the matrix notation, but who nevertheless may be interested in new applications of CFR. We surmise that the best notation in each case is a matter of personal taste, and that the important point is that all mathematical steps remain clear.

**Ln 192: Residual term with what assumed properties?**
**A:** The assumed properties are of white or red noise. We have included now in the revised text.

**Paragraph starting on Ln 279: This paragraph is full of undefined jargon that is not cited. It is meaningless for the uninitiated. Please correct.**
**A:** This paragraph has been revised for clarity, adding cfurther itations (Knerr, et, al. 1990; Yu, et, al. 2019) and removing the amount of jargon as much as possible.

**Ln 299: This was first noted in Smerdon et al. (2010, 2011).**
**A:** Noted and corrected

**Ln 310: "reduction in skill" as opposed the vague use of degradation here?**

**A:** Noted and corrected

**Ln 320: In the spirit of my general comments, one curiosity is why CCA does not perform better than PCR with regard to the cc metric. CCA is designed to optimize the correlation, which is why it can sometimes yield larger variance losses. It is therefore curious why it doesn't universally beat out PCR in the cc metric.**
**A:** CCA indeed maximizes the correlation that can be attained with a change of variables, i.e. with a linear combination of the grid-cell series in each of the two fields. In an admittedly artificial example, the canonical correlation can be very high and yet the reconstruction skill in general can remain low. If one grid cell in each field are very highly correlated to one another (and assuming here no PCA pre-filtering), CCA will pick those two cells as the first CCA pair (i.e., a pattern in each field with very high loadings on those cells). The rest of the cells will not contribute to the CCA pattern. The reconstruction skill will therefore be very low in all those cells, despite the CCA being very high. In general, the reconstruction skill will be a monotonic function of the canonical correlation coefficient and the variance explained by the canonical predictand pattern. If the latter is low, the reconstruction skill will be low in large areas of the predictand field.
We have included a few sentences to guide the reader on this point.

**Ln 327: The variance losses have a relatively straightforward interpretation for the traditional regression approaches. When analyzing the mean results, the variance losses reflect loss of signal (reflected in the mean) and increases in the variance associated with the error term. It would be useful to know if the machine learning method can be interpreted in the same way, or if there is an alternative way to think about variance losses for that method.**
**A:** We believe that in the case of machine learning algorithms, but the interpretation may be more difficult. We also point out here that, even in the reviewer's interpretation, simple algorithms can yield contrasting results. For instance, even assuming that noise is present in the predictand only, direct linear regression and inverse linear regression lead to under or over estimation of the reconstructed variance, respectively. In the case of neural network, it is very common that they also lead to underestimation of the variance (noted already long time ago, Zorita et al., 1999 https://doi.org/10.1175/1520-0442(1999)012%3C2474:TAMAAS%3E2.0.CO;2), but this can depend on the structure of the network and specifically on the form of the prescribed neuron activation function.

**Ln 390: Why should complexity translate to improved skill? I am aware of no grand postulate that makes this case.**
**A:** The reviewer is correct that there is no general principle linking complexity and skill, but it is reasonable to assume that a more complex model might be able to better capture more complex relationships. For instance, a linear model cannot capture non-linear links outside a narrow range. Artificial neural network is a subset of machine learning method could usually be understood as a universal approximator which can map and approximate any kind of linear or nonlinear functions by selecting a suitable set of connecting weights and transfer functions in principle (Hornik et al., 1989). Thus, it is reasonable to assume that a better performance might be achieved, but indeed, this not generally guaranteed. We have reformulated that sentence

**Ln 402: The relationship (or lack thereof) between the skill of the mean indices and spatial skill was first discussed in Smerdon et al. (2010, 2011) and further highlighted in Smerdon's 2012 pseudoproxy review.**
**A:** Noted and corrected

**Ln 418: This is vague. What about alternative methods might be useful in the context of the CFR problem? There are lots of methods out there. What direction can the authors provide, based on the work they have done, that might represent useful characteristics in other machine-learning methods to try in the context of this problem?**

**A:** We now briefly present alternative directions and methods that might be useful in paleo CFR experiments based on more realistic calibration and validation time period as the reviewer suggested. Our first implementation of the more complex Bi-LSTM does not show superiority in CFRs compared to traditional CFR methods, so we would like to draw an assumption that more complicated architecture might not be helpful for CFRs at least based on our specific experiment results and the employed architecture of Bi-LSTM. However, we would suggest an Echo State Network (ESN) for paleo climate research. (Lukosevicius, M. & Jaeger, H. 2009; Nadiga, 2020). Both ESN and LSTM belong to RNN, yet ESN is much simpler than LSTM (Lukosevicius, M. & Jaeger, H. 2009), and also has outperformed the RNN methods in other applications (Chattopadhyay et al., 2019, Nadiga, B. 2020). We thus encourage testing ESN in different paleo climate research directions.

*Ln 451: CCA is a classic linear-based CFR method. This structure is awkward.*
**A:** Noted and corrected

**Ln 460: "Reservoir Computing methods-Echo State Network" is screaming for a reference so that the rest of us can figure out what it is.**

**A:** We have included references on ESN: e.g., Lukosevicius, M. & Jaeger, H. 2009 and Nadiga, 2020

**Figures 6 and 7: Much of the text in this figure would only be legible by Ant Man. I strongly suggest increasing the size of the legend, fonts, and axis labels.**

**A:** Noted and corrected

**Figures 8 and 9: I find these figures very hard to read. Why include the bar plots for the data bins? It would be much clearer to simply show the estimated PDFs, which characterize the behavior well enough.**

A: The reason for including the bin bars is to highlight the differences in the frequency of extremes. The smoothed PDFs can provide a picture that is not totally accurate. We have now included figures that show the smoothed PDFs and, additionally, a few bin bars for extreme events

**References:**
Bengio, Y., Simard, P. & Frasconi, P. Learning long-term dependencies with gradient descent is difficult. IEEE Trans. Neural Networks 5, 157–166 (1994).
Hochreiter, S. & Schmidhuber, J. Long short-term memory. Neural Comput. 9, 1735–1780 (1997). This paper introduced LSTM recurrent networks, which have become a crucial ingredient in recent advances with recurrent networks because they are good at learning long-range dependencies.
Hornik, K, Stinchcombe, M, White, H. Multilayer feedforward networks are universal approximators. Neural Networks, 2 (1989), pp. 359-366.

Rasp, S. & Lerch, S. Neural networks for postprocessing ensemble weather forecasts. Mon. Weather Rev. 146, 3885–3900 (2018).

Castelvecchi, D. Can we open the black box of AI? Nature 538, 20–23 (2016).

Toms, B. A., Barnes, E. A., & Hurrell, J. W. (2021). Assessing decadal predictability in an Earth-system model using explainable neural networks. Geophysical Research Letters, 48, e2021GL093842. https://doi.org/10.1029/2021GL093842

Molnar, C. Interpretable Machine Learning—A Guide for Making Black Box Models Explainable. 2019. Available online: https://christophm.github.io/interpretable-ml-book/

Christiansen, B.: Reconstructing the NH mean temperature: can underestimation of trends and variability be avoided?, J. Clim., 24, 674–692, 2011.

Knerr, S., Personnaz, L and Dreyfus, G., "Single-layer learning revisited: A stepwise procedure for building and training a neural network" in Neurocomputing: Algorithms Architectures and Applications, Berlin, Germany:Springer, pp. 41-50, 1990, [online] Available: http://dx.doi.org/10.1007/978-3-642-76153-9-5.

Yu, Y., Si, X., Hu, C. & Zhang, J. A review of recurrent neural networks: LSTM cells and network architectures. Neural Comput. 31, 1–36 (2019).

Lukoševičius, M. & Jaeger, H. Reservoir computing approaches to recurrent neural network training. Computer Science Review 3, 127–149 (2009).

Nadiga, B.: Reservoir Computing as a Tool for Climate Predictability Studies, J. Adv. Model. Earth Sy., e2020MS002290, https://doi.org/10.1029/2020MS002290, 2020.

Chattopadhyay, A., Hassanzadeh, P., and Subramanian, D.: Data-driven predictions of a multiscale Lorenz 96 chaotic system using machine-learning methods: reservoir computing, artificial neural network, and long short-term memory network, Nonlin. Processes Geophys., 27, 373–389, 2020.

---

## Author Response (AR2)

We would like to thank Referee 1 for their detailed and constructive comments. In the following to explain how we have modified the manuscript to address their suggestions.

**The original reviewer's suggestions are written in bold font** and our responses with normal font.

**General comments**
**I want to thank the authors for addressing many of my concerns and I commend them for more thoroughly investigating the LSTMs architecture and hyperparameters to try and get the LSTMs to work. I think the new discussion at the end of the paper is helpful for understanding why the LSTM's are fundamentally failing to improve over linear methods, i.e. the implied link is linear. I think that finding, and others from section 4.1, are important enough to warrant inclusion in the abstract since they're broader claims about non-linear methods for CFR that this paper found evidence for. Could the bi-LSTM then be seen as more of a tool for investigating the (non)amenability of CFR to non-linear methods in small data regimes, rather than just an "off the shelf" application that didn't pan out? If that understanding is correct, then I would consider emphasizing that a little more up front.**
A: We think that our original major research scope about this manuscript is for the purpose of testing and verifying our working hypothesis that whether a more complex machine-learning method LSTM would provide better reconstructions for temperature field. At the meantime, we noticed that in this context, we were using a limited amount of dataset to train and validate the neural network method, which could be, on the other side, an investigation of neural network performance as a tool for CFRs especially when only a limited data is available or employed compared to 'big data' and the usually big data-drive deep neural network aspect. We have add sentences into this context to mention this.

**Specific comments**
1. **Line 112-113 — "However, the prior use of information from climate models precludes a posterior critical comparison between simulations and reconstructions, and thereby the resulting reconstructions lose one appeal of climate reconstructions in general." — I do not understand what this is trying to say. Can you please explain this again? Actually I'm struggling to understand the overall reasoning behind DA not being applicable here. DA was compared to PCA in (Steiger et al., 2014) and they claimed their method was computationally efficient. Is there something fundamentally different about the approach presented here that makes DA not applicable**?
A: We wanted to raise two different points regarding DA methods. One is that the DA methods use a lot of information stemming from simulations with climate models. Therefore, the a posteriori comparison of DA-based reconstructions and climate simulations is compromised, and both data sets are not independent. The second point it is difficult to methodologically evaluate DA-methods against other purely statistical methods, since the former use much more information and data (from climate simulations), and thus the comparison cannot be fair.
We have reformulated this paragraph to make these two points more clear.

2. **Table 1 — Why do PCR and CCA improve in the noisy scenarios but LSTM deteriorates**?
A: We believe that maybe because these noise-contaminated data cause obvious overestimations in the amplitude of reconstructed variability for the linear PCA and CCA methods. Some noise signal may deteriorates the reconstructions, while these noise single may also lead to good reconstructions, since the CFR reconstructions are effected by many factors, such as the proxy numbers and its spatial distributions, random noise signal introduced and added to certain important spatial proxy locations which could have significant effect on the overall spatial reconstruction may result in a general better reconstructions. For the nonlinear machine learning methods, it is very sensitive to external noise. Kalapanidas et al., 2003 and Atla

A, et al., 2011, demonstrated that linear regression can perform better results than nonlinear methods considering noise sensitivity studies. And some studies indicated that external interference or noise could damage the ability of neural networks (Heaven 2019).

3. **Line 527: "PCR and CCA exhibit overestimated reconstructions within noisy PPEs, the Bi-LSTM presents relatively robust reconstructions" — What are overestimated reconstructions? Is this indicating some advantage of LSTM**?

A: The overestimated reconstruction here refers to the overestimation in the amplitude of variability. The overestimation is represented by the ratio of standard deviations, as a metric for assessing the reconstruction variance. A SD ratio close to 1 indicates we achieve perfect reconstructions with the same amplitude as the target. In these noisy PPEs, the linear regression method we employed, especially the PCR method, exhibits obvious overestimations (the value of SD ratio is bigger than 1 over some spatial regions as shown in Fig 6-7). In principle, it is difficult for regression methods to reproduce perfect reconstructions (obtaining SD ratio equals 1).

The LSTM method shows relatively robust reconstructions within these noisy PPEs. However, as we explained in the above comments, neural network is also very sensitive in noisy experiments. In our CFR study, we employed a small sample size and add two type of noise to contaminate these original data. We would conclude that the CFR results based on neural network would be much more dependent on the different scenarios. Since several external factors, such as data set and noise type, and internal factors such as interpretability of neural network, would have significant on drawing a general conclusion about the final reconstructions. But based on our experiments, the LSTM architecture tested in our study seems to show some advantage in achieving reasonably robust reconstructions.

4. **Line 555: "Both ESN and LSTM belong to the family of RNN, yet ESN is much simpler than LSTM (Lukosevicius and Jaeger 2009), and has outperformed the RNN methods in other applications (Chattopadhyay et al., 2019; Nadiga, B. 2020)." — If ESN's are simpler and more promising, then why does this paper stick to the LSTM model, which clearly did not improve over simpler methods, and not just propose and evaluate ESN's, even if alongside the LSTM**?

A: The results regarding the LSTM have been collected along this study, and this experience lead us to think that the ESM could be more promising. This assumption is based on a few preliminary results, but not on a through testing. However, we cannot be sure at this stage that this will turn out to be correct. Our plan is to test the ESM in a follow-up publication.

The ESN method we mentioned in this manuscript is because we have already implemented further CFR experiments by employing this ESN and also compared it with the LSTM method.

Refereces:

Kalapanidas E, Avouris N, Craciun M, Neagu D. (2003). Machine learning algorithms: a study on noise sensitivity. In Proc. 1st Balcan Conference in Informatics. pp. 356–365.

Atla A, et al. Sensitivity of different machine learning algorithms to noise. J Comput Sci Coll. 2011;26(5):96–103.

Heaven, D. Why deep-learning AIs are so easy to fool. Nature 574, 163–166 (2019).

We would like to thank Referee 2 for their detailed and constructive comments. In the following to explain how we have modified the manuscript to address their suggestions.

**The original reviewer's suggestions are written in bold font** and our responses with normal font.

**General comments**
**The authors have done a good job of revising the manuscript in response to my original review. The article is now more comprehensive, contextualized, and described. I support publication after the items I list below are addressed. As a side note, I do not point out many of the typos and grammatical errors, but the paper would benefit from detailed language editing**.
A: We have iterated the manuscript on correcting typos and grammatical errors.

**Specific Comments:**
**Ln. 14: field,.**
A: we have corrected this typo.

**Ln 117: My point about the AMV is that the observational AMV is not defined exclusively as the "decadal filtered surface temperature anomaly." It is not even necessarily the decadally filtered anomaly, but the index of average north Atlantic SSTs after removing the forced signal in that average (whether by removal of a linear trend or otherwise). I am not suggesting that the authors use a different definition to isolate the AMV in the longer last-millennium runs, but to better define the AMV in this location**.
A: As the reviewer points out, this is a terminology issue, the use of which is not quite clear through the literature. For instance, the GFSL site on the AMV defines the purely internal variability of the North Atlantic SST as Atlantic Multidecadal Oscillation (AMO) whereas the AMV will include natural and externally forced variability. Other authors indeed refer with AMV to the internal variability only. The new version is now more specific on this terminology.

**Ln 127-150: This is useful information and should be included in the Data and Methods sections. This is nevertheless a strange collection of information to include in the Intro. The authors may want to summarize some of this information as part of a roadmap in the Intro, but much of it should be incorporated into the Data and Methods sections (which does not include some of this important info, i.e. it is not just repeated here)**.
A: We agree with this comment, and have made changes correspondingly, moved the proxy and climate model information from Introduction to Data and Method section.

**Ln 172: The use of CESM1 and CAM5 is strangely garbled here and elsewhere in the manuscript. In all cases that the authors are talking about the CESM-LME results they should refer to CESM. CAM5 is the atmospheric model used in the CESM1 coupled model. Use of CESM1-CAM5 is strange (CESM1 by definition uses CAM5 in its architecture), while all of the figures identify the CESM-LME results as CAM5. This should all be remedied**.
A: We have corrected all the CESM1, CAM5 and CESM1-CAM5 into CESM in the text, and corrected all CAM/CAM5 caption into CESM in all figures.

**Ln 180: "We use ensemble member 13 from the CESM-LME as the basis for our CESM pseudoproxy experiments." Note also that LME is defined in this sentence but it is used in line 174 without definition**.

A: We have corrected this.

**Ln 378-9: This does indeed support the stationarity of the teleconnection patterns, but also says something about the physical nature of the patterns, i.e. they are to some degree localized and do not share significant amounts of covariance outside of the regions where they are sampled**.
A:

**Ln 409: This is only true if the EOFs in the training interval are stationary, i.e. well represent the EOF patterns in the reconstruction interval as well**.
A: In our manuscript, we have indicated that we assume the EOF patterns derived from training interval remain constant in time, which is stationary with time.

**Ln 424: It is not clear whether these assessments were done for EOF patterns over the reconstruction interval, the training interval, or both. Interpretation of the results depends on this choice, namely whether stationarity is part of what is being assessed. Reduced skill in the recon-interval EOFs could be both associated with deficiencies in the methods or non-stationarity in the EOF structure. This should be made more clear**.
A: We corrected this. Here in Line 424 the EOF patterns were derived from the reconstruction interval.

**Ln 434: The stated explanation of the Yun et al. methodological choice and its potential statistical artifacts is not clear here. The few sentences that start here should be more clearly articulated**.
A: We integrated an additional paragraph to articulate this more clearly.

**Ln 454: I believe Figure 8 should be Figure 11 here.**
A: We have corrected this typo.

**Ln 466: I believe Figure 11 should be Figure 12 here.**
A: We have corrected this typo

**Ln 482: There is a general and non-quantitative discussion that starts here about how one distribution describes the target distribution "better." This is generally vague language that begs to be quantified. The KS tests are the quantitative part of this discussion and are more sufficient for describing things as better or worse. I would combine the language, or just move directly to the KS tests as a means of characterizing how well the distributions compare**.
A: We think that a detailed text description of histogram may be necessary for people to better distinguish the capability of each different method on capturing the extremes – lower or upper tails intuitively. The KS statistic would be then cable of better describing the detailed differences of each reconstruction methods in a quantitatively way. Considering the advice of the reviewer, we have changed the ordering of these two paragraphs. Now the quantitative results derived from the KS tests are presented first, followed by our more qualitative description of the behavior of the distributions at their tails The reader encounters first the quantitative tests and then better appreciate the more subtle differences a the fringes of the distribution.

**Ln 527: There are multiple places in the discussion where the authors use overestimated reconstructions or similar constructs. I think what they mean is overestimated variance, which should be used to be precise**.
A: We have corrected the overestimated related reconstructions to overestimated variance.

---

## Author Response (AR3)

We would like to thank the Editor for his detailed comments. In the following we explain how we have modified the manuscript to address his suggestions. All changes are highlighted in **yellow** front in the revised manuscript.

**The original suggestions are written in bold font** and our responses with normal font.

**Referee #1 / General comments #2, #3 and #4 and Referee #2 / line 409: You have responded to these comments, but you do not appear to have made any relevant changes to the manuscript. If you have revised the manuscript, please highlight the changes that you have made. If you have not revised the manuscript, then please consider adding at least some of the information that you provide in your responses.**

*Referee 1 General comments:*
*I want to thank the authors for addressing many of my concerns and I commend them for more thoroughly investigating the LSTMs architecture and hyperparameters to try and get the LSTMs to work. I think the new discussion at the end of the paper is helpful for understanding why the LSTM's are fundamentally failing to improve over linear methods, i.e. the implied link is linear. I think that finding, and others from section 4.1, are important enough to warrant inclusion in the abstract since they're broader claims about non-linear methods for CFR that this paper found evidence for. Could the bi-LSTM then be seen as more of a tool for investigating the (non)amenability of CFR to non-linear methods in small data regimes, rather than just an "off the shelf" application that didn't pan out? If that understanding is correct, then I would consider emphasizing that a little more up front*
A: In response to this general comment, we elaborated a little more than in our original version that an important objective of our study testing and verifying our working hypothesis that a more complex machine-learning method LSTM would provide better temperature reconstructions with the right amplitude of variations. We noticed that in this context, we were bound to use a limited amount of data to train the machine-learning method compared to other 'big data' applications.
In the revised manuscript, we have included in the abstract a summary of the new discussion contents in section 4.1. We have also added a sentence into the abstract that the Bi-LSTM could be a tool for exploring the amenability of CFRs especially in small data regimes.

*2. Table 1 — Why do PCR and CCA improve in the noisy scenarios but LSTM deteriorates?*
A: For the nonlinear machine learning methods, it is very sensitive to external noise. Kalapanidas et al., 2003 and Atla A, et al., 2011, demonstrated that linear regression can perform better results than nonlinear methods considering noise sensitivity studies. And some studies indicated that external interference or noise could damage the ability of neural networks (Heaven 2019). This is explained in lines 536.
We have also added sentence to elaborate more on the performance explanation in the Discussion section of the revised manuscript.

*3. Line 527: "PCR and CCA exhibit overestimated reconstructions within noisy PPEs, the Bi-LSTM presents relatively robust reconstructions" — What are overestimated reconstructions? Is this indicating some advantage of LSTM?*
A: The overestimated reconstruction here refers to the overestimation in the amplitude of variability. The overestimation is represented by the ratio of standard deviations, as a metric for assessing the reconstruction variance. A  SD ratio close to 1 indicates we achieve perfect reconstructions with the same amplitude as the target. In these noisy PPEs, the linear regression method we employed, especially the PCR method, exhibits obvious overestimations (the value of SD ratio is bigger than 1 over some spatial regions as shown

in Fig 6-7). This comment is related to the previous comment, and we believe it is also addressed in lines 536.

We have added sentence to indicate that the overestimated reconstructions refer to the reconstructions in the amplitude of climatic variability. And added sentence to indicate the possible advantage of LSTM in CFRs.

*4. Line 555: "Both ESN and LSTM belong to the family of RNN, yet ESN is much simpler than LSTM (Lukosevicius and Jaeger 2009), and has outperformed the RNN methods in other applications (Chattopadhyay et al., 2019; Nadiga, B. 2020)." — If ESN's are simpler and more promising, then why does this paper stick to the LSTM model, which clearly did not improve over simpler methods, and not just propose and evaluate ESN's, even if alongside the LSTM?*

A: The results regarding the LSTM have been achieved along this study, and this experience lead us to think that the ESM could be more promising. This assumption is based on a few preliminary results, but not on a thorough testing. However, we cannot be sure at this stage that this will turn out to be correct. Our plan is to test the ESM in a follow-up publication.

The ESN method we mentioned in this manuscript is because we have already implemented further CFR experiments by employing this ESN and also compared it with the LSTM method. This is explained in lines 565.

We have added a sentence to emphasize that we will implement further steps to test ESN method in CFRs.

Refereces:
Kalapanidas E, Avouris N, Craciun M, Neagu D. (2003). Machine learning algorithms: a study on noise sensitivity. In Proc. 1st Balcan Conference in Informatics. pp. 356–365.
Atla A, et al. Sensitivity of different machine learning algorithms to noise. J Comput Sci Coll. 2011;26(5):96–103.
Heaven, D. Why deep-learning AIs are so easy to fool. Nature 574, 163–166 (2019).

**Referee #2 / Lines 378-379: You have not responded to this comment. Please do so.**
*Ln 378-9: This does indeed support the stationarity of the teleconnection patterns, but also says something about the physical nature of the patterns, i.e. they are to some degree localized and do not share significant amounts of covariance outside of the regions where they are sampled.*

A: We agree with this comment, and have included this suggestion into the revised manuscript. The changes are highlighted in yellow font in lines 380.

*Ln 409: This is only true if the EOFs in the training interval are stationary, i.e. well represent the EOF patterns in the reconstruction interval as well.*

A: In our manuscript, we have indicated that the EOF patterns derived from training interval are assumed to remain constant in time. This is also an assumption that made in real paleo reconstructions, otherwise the PCR method would not be valid. This stated in lines 410.

We have added sentence to state that the dominant EOF patterns are assumed to be stationary/constant with time in our PPEs.

**Technical corrections:**
*Line 24-25: Please replace "simple" with "samples" and "dataset" with "datasets". I'm also not sure what you mean by "positive tests". Perhaps you could replace "positive tests on" with "that skill can be achieved even when" or similar.*

A: We have corrected it in lines 25.